# Global Convergence of Least Squares EM
# for Demixing Two Log-Concave Densities

**Wei Qian, Yuqian Zhang, Yudong Chen**
School of Operations Research and Information Engineering
Cornell University
{wq34,yz2557,yudong.chen}@cornell.edu

## Abstract

This work studies the location estimation problem for a mixture of two rotation invariant log-concave densities. We demonstrate that Least Squares EM, a variant of the EM algorithm, converges to the true location parameter from a randomly initialized point. Moreover, we establish the explicit convergence rates and sample complexity bounds, revealing their dependence on the signal-to-noise ratio and the tail property of the log-concave distributions. Our analysis generalizes previous techniques for proving the convergence results of Gaussian mixtures, and highlights that an angle-decreasing property is sufficient for establishing global convergence for Least Squares EM.

## 1 Introduction

One important problem in statistics and machine learning [18, 24] is learning a finite mixture of distributions. In the parametric setting in which the functional form of the density is known, this problem is to estimate parameters (e.g., mean and covariance) that specify the distribution of each mixture component. The parameter estimation problem for mixture models is inherently non-convex, posing challenges for both computation and analysis. While many algorithms have been proposed, rigorous performance guarantees are often elusive. One exception is the Gaussian Mixture Model (GMM) for which much theoretical progress has been made in recent years. The goal of this paper is to study algorithmic guarantees for a much broader class of mixture models, namely the mixture of log-concave distributions. This class includes may common distributions[1] and is interesting from both modelling and theoretical perspectives [2, 3, 6, 12, 26, 23].

We focus on the Expectation Maximization (EM) algorithm [11], which is one of the most popular methods for estimating mixture models. Understanding the convergence property of EM is highly non-trivial due to the non-convexity of the negative log-likelihood function. The work in [4] developed a general framework for establishing *local* convergence to the true parameter. Proving *global* convergence of EM is more challenging, even in the simplest setting with a mixture of two Gaussians (2GMM). The recent work in [10, 28] considered balanced 2GMM with known covariance matrix and showed for the first time that EM converges to the true location parameter using random initialization. Subsequent work established global convergence results for a mixture of two truncated Gaussians [19], two linear regressions (2MLR) [17, 16], and two one-dimensional Laplace distributions [5].

All the above results (with the exception of [5]) rely on the explicit density form and the specific properties of the Gaussian distribution. In particular, under the the Gaussian assumption, the M-step in the EM algorithm has a closed-form expression, which allows a direct analysis of the convergence behavior of the algorithm. However, for a general log-concave distribution the M-step no longer admits a closed-form solution, and this poses significant challenges for analysis. To address

this difficulty, we consider a modification of the standard EM algorithm, called *Least Squares EM* (LS-EM), to learn the location parameter of a mixture of two log-concave distributions. This algorithm admits an explicit update, which is computationally simple.

As the main result of this paper, we show that for a mixture of rotation invariant log-concave distribution, LS-EM converges to the true location parameter with a random initialization. Moreover, we provide explicit convergence rates and sample complexity bounds, which depend on the signal-to-noise ratio as well as the tail property of the distribution. As the functional form of the true density may be unknown, we further establish a robustness property of LS-EM when using a mis-specified density. As a special case, we show that using a Gaussian density, LS-EM globally converges to a solution close to the true parameter whenever the variance of the true log-concave density is moderate.

**Technical Contributions**  We generalize the sensitivity analysis in [10] to a broad class of log-concave distributions. In the process, we demonstrate that log-concavity and rotation invariance of the distribution are the only properties required to guarantee the global convergence of LS-EM. Moreover, our analysis highlights the fundamental role of an *angle-decreasing* property in establishing the convergence of LS-EM to the true location parameter in the high dimension settings. Note the $\ell_2$ distance contraction, upon which the previous convergence results were built, no longer holds for general log-concave distributions.

**Organization**  In Section 2, we describe the parameter estimation problem for a mixture of log-concave distributions and review related work. In Section 3, we derive the LS-EM algorithm and elucidate its connection with classical EM. Analysis of the global convergence of LS-EM is provided in Section 4 for the population setting, in Section 5 for the finite-sample setting, and in Section 6 for the model mis-specification setting. The paper concludes with a discussion of future directions in Section 7. Some details of the proofs and numerical experiments are deferred to the Appendix.

**Notations**  We use $x \in \mathbb{R}$ and $\boldsymbol{x} \in \mathbb{R}^d$ to denote scalars and vectors, respectively; $X \in \mathbb{R}$ and $\boldsymbol{X} \in \mathbb{R}^d$ to denote scalar and vector random variables, respectively. The $i$-th coordinate of $\boldsymbol{x}$ (or $\boldsymbol{X}$) is $x_i$ (or $X_i$), and the $j$-th data point is denoted by $\boldsymbol{x}^j$ or $\boldsymbol{X}^j$. The Euclidean norm in $\mathbb{R}^d$ is $\|\cdot\|_2$. For two vectors $\boldsymbol{\alpha}, \boldsymbol{\beta} \in \mathbb{R}^d$, we use $\angle(\boldsymbol{\alpha}, \boldsymbol{\beta}) \in (0, \pi)$ to denote the angle between them and $\langle \boldsymbol{\alpha}, \boldsymbol{\beta} \rangle$ to denote their inner product. Finally, $\boldsymbol{I}_d$ is the $d$-by-$d$ identity matrix.

## 2  Problem Setup

In this section, we set up the model for a mixture of log-concave distributions and discuss the corresponding location estimation problem.

### 2.1  Data Generating Model

Let $\mathcal{F}$ be a class of rotation invariant log-concave densities in $\mathbb{R}^d$ defined as follows:

$$\mathcal{F} = \left\{ f : f(\boldsymbol{x}) = \frac{1}{C_g} \exp\left(-g(\|\boldsymbol{x}\|_2)\right), g \text{ is convex and strictly increasing on } [0, \infty), \right.$$
$$\left. \int f(\boldsymbol{x}) \, d\boldsymbol{x} = 1, \int x_i^2 f(\boldsymbol{x}) \, d\boldsymbol{x} = 1, \forall i \in [d] \right\}, \tag{1}$$

where we may assume $g(0) = 0$ without loss of generality.[2] It can be verified that each $f \in \mathcal{F}$ has mean $\boldsymbol{0}$ and covariance matrix $\boldsymbol{I}_d$. For each $f \in \mathcal{F}$, we may generate a location-scale family consisting of the densities $f_{\boldsymbol{\beta}, \sigma}(\boldsymbol{x}) := \frac{1}{\sigma^d} f\left(\frac{\boldsymbol{x} - \boldsymbol{\beta}}{\sigma}\right)$, which has mean $\boldsymbol{\beta}$ and covariance matrix $\sigma^2 \boldsymbol{I}_d$.

We assume that each data point is generated from $D(\boldsymbol{\beta}^*, \sigma)$, a balanced mixture of two symmetric members of the above log-concave location-scale family:

$$D(\boldsymbol{\beta}^*, \sigma) := \frac{1}{2} f_{\boldsymbol{\beta}^*, \sigma} + \frac{1}{2} f_{-\boldsymbol{\beta}^*, \sigma}. \tag{2}$$

Throughout this paper, we denote the signal-to-noise ratio (SNR) by

$$\eta := \|\boldsymbol{\beta}^*\|_2/\sigma.$$

It is sometimes useful to view the above model as an equivalent latent variable model: for each $i \in [n]$ an unobserved label $Z_i \in \{1, 2\}$ is first generated according to $\mathbb{P}(Z = 1) = \mathbb{P}(Z = 2) = 1/2$, and then the data point $\boldsymbol{X}^i$ is sampled from the corresponding mixture component, i.e., from $f_{\boldsymbol{\beta}^*,\sigma}$ if $Z_i = 1$ and from $f_{-\boldsymbol{\beta}^*,\sigma}$ otherwise.

**Examples:** Below are several familiar examples of one-dimensional log-concave distributions $f \propto \exp(-g)$ from $\mathcal{F}$:

1. *Polynomial distributions*: $g(x) \propto |x|^r$ with $r \geq 1$. When $r = 2$, it corresponds to the *Gaussian* distribution. When $r = 1$, it corresponds to the *Laplace* distribution.

2. *Logistic distribution*: $g(x) \propto \log(e^{-|x|/2} + e^{|x|/2})$.

These distributions can be generalized to higher dimension by replacing $|x|$ with $\|\boldsymbol{x}\|_2$. In Appendix B, we provide a review of some elementary properties of log-concave distributions.

## 2.2 Location Estimation and the EM Algorithm

We assume that $\sigma$ is known, and our goal is to estimate the location parameter $\boldsymbol{\beta}^*$ given data $\boldsymbol{X}^1, \boldsymbol{X}^2, \ldots, \boldsymbol{X}^n \in \mathbb{R}^d$ sampled i.i.d. from the mixture distribution $D(\boldsymbol{\beta}^*, \sigma)$ in (2). We first consider this problem for a given log-concave family for which the base density $f$ (equivalently, $g$) is known. The case with an unknown $f$ is discussed in Section 6.

Since the negative log-likelihood function of the mixture (2) is non-convex, computing the standard MLE for $\boldsymbol{\beta}^*$ involves a non-convex optimization problem. EM is a popular iterative method for computing the MLE, consisting of an expectation (E) step and a maximization (M) step. In a standard implementation of EM, the E-step computes the conditional distribution of the labels $Z_i$ under the current estimate of $\boldsymbol{\beta}^*$, and the M-step computes a new estimate by maximizing the expected log-likelihood based on the E-step. The LS-EM algorithm we consider, described in Section 3 to follow, is a variant of the standard EM algorithm with a modified M-step.

## 2.3 Convergence of EM and Related Work

Despite the popularity and empirical success of the EM algorithm, our understanding of its theoretical property is far from complete. Due to the non-convexity of negative log-likelihood functions, EM is only guaranteed to converge to a stationary point [27]. Quantitative convergence results only began to emerge in recent years. The work [4] proposed a general framework for establishing the *local* convergence of EM when initialized near the true parameter, with applications to 2GMM, 2MLR, and regression with missing coefficients. Extensions to multiple components are considered in [29].

Beyond local convergence, it is known that the likelihood function of GMM may have bad local optima when there are more than two components, and EM fails to find the true parameter without a careful initialization [14]. Analysis of the global convergence of EM has hence been focused on the two component setting, as is done in this paper. The work in [10] showed that EM converges from a random initialization for 2GMM. Subsequent work in [17, 16, 28, 19] established similar results in other settings, most of which involve Gaussian models. An exception is [5], which proved the global convergence of EM for a mixture of 2 Laplace distributions and derived an explicit convergence rate, but only in the one-dimensional population (infinite sample) setting. In general, properties of EM for mixtures of other distributions are much less understood, which is the problem we target at in this paper.

The log-concave family we consider is a natural and flexible generalization of Gaussian. This family includes many common distributions, and has broad applications in economics [2, 3], reliability theory [6] and sampling analysis [12]; see [26, 23] for a further review. Existing work on estimating log-concave distributions and mixtures has mostly considered the non-parametric setting [26, 9, 15, 21, 9]; these methods are flexible but typically more computational and data intensive than the parameteric approach we consider. Other approaches of learning general mixtures include spectral [1, 22] and tensor methods [13, 8]; EM is often applied to the output of these methods.

# 3 The Least Squares EM Algorithm (LS-EM)

The M-step in the standard EM involves maximizing the conditional log-likelihood. For GMM, the M-step is equivalent to solving a least-squares problem. For a mixture of log-concave distributions, the M-step is to solve a convex optimization problem, and this optimization problem does not admit a closed form solution in general. This introduces complexity in both computation and analysis.

We instead consider Least Squares EM, a variant of EM that solves a least-squares problem in the M-step even for non-Gaussian mixtures. To elucidate the algorithmic property, we consider LS-EM in the population setting, where we have access to an infinite number of data points from the mixture distribution $D(\boldsymbol{\beta}^*, \sigma)$. The finite sample version is discussed in Section 5.

Each iteration of the population LS-EM algorithm consists of the following two steps:

- **E-step:** Compute the conditional probabilities of the label $Z \in \{1, 2\}$ given the current location estimate $\boldsymbol{\beta}$:

$$p_{\boldsymbol{\beta},\sigma}^1(\boldsymbol{X}) := \frac{f_{\boldsymbol{\beta},\sigma}(\boldsymbol{X})}{f_{\boldsymbol{\beta},\sigma}(\boldsymbol{X}) + f_{-\boldsymbol{\beta},\sigma}(\boldsymbol{X})}, \qquad p_{\boldsymbol{\beta},\sigma}^2(\boldsymbol{X}) := \frac{f_{-\boldsymbol{\beta},\sigma}(\boldsymbol{X})}{f_{\boldsymbol{\beta},\sigma}(\boldsymbol{X}) + f_{-\boldsymbol{\beta},\sigma}(\boldsymbol{X})}. \qquad (3)$$

- **Least-squares M-step:** Update the location estimate $\boldsymbol{\beta}$ via weighted least squares:

$$\boldsymbol{\beta}^+ = \underset{\boldsymbol{b}}{\arg\min} \, \mathbb{E}_{\boldsymbol{X} \sim D(\boldsymbol{\beta}^*,\sigma)} \left[ p_{\boldsymbol{\beta},\sigma}^1(\boldsymbol{X}) \|\boldsymbol{X} - \boldsymbol{b}\|_2^2 + p_{\boldsymbol{\beta},\sigma}^2(\boldsymbol{X}) \|\boldsymbol{X} + \boldsymbol{b}\|_2^2 \right]$$

$$= \mathbb{E}_{\boldsymbol{X} \sim D(\boldsymbol{\beta}^*,\sigma)} \boldsymbol{X} \tanh \left( \frac{1}{2} g \left( \frac{1}{\sigma} \|\boldsymbol{X} + \boldsymbol{\beta}\|_2 \right) - \frac{1}{2} g \left( \frac{1}{\sigma} \|\boldsymbol{X} - \boldsymbol{\beta}\|_2 \right) \right) := M(\boldsymbol{\beta}^*, \boldsymbol{\beta}). \qquad (4)$$

In (4), we minimize the sum of squared distances of $\boldsymbol{X}$ to each component's location, weighted by the conditional probability of $\boldsymbol{X}$ belonging to that component. One may interpret LS-EM as a soft version of the K-means algorithm: instead of associating each $\boldsymbol{X}$ exclusively with one of the components, we assign a probability, which is computed using the log-concave density.

## 3.1 Connection to Standard EM

In contrast to LS-EM, the M-step in the standard EM algorithm involves maximizing the weighted log-likelihood function (or minimizing the weighted negative log-likelihood function):

**Standard M-step:**
$$\underset{\boldsymbol{b}}{\arg\max} \, Q(\boldsymbol{b} \mid \boldsymbol{\beta}) := \mathbb{E}_{\boldsymbol{X} \sim D(\boldsymbol{\beta}^*,\sigma)} \left[ p_{\boldsymbol{\beta},\sigma}^1(\boldsymbol{X}) \log f_{\boldsymbol{b},\sigma}(\boldsymbol{X}) + p_{\boldsymbol{\beta},\sigma}^2(\boldsymbol{X}) \log f_{-\boldsymbol{b},\sigma}(\boldsymbol{X}) \right]. \qquad (5)$$

The standard EM iteration, consisting of (3) and (5), corresponds to a minorization-maximization procedure for finding the MLE under the statistical setting (2). In particular, the function $Q(\cdot \mid \boldsymbol{\beta})$ above is a lower bound of the (marginal) log-likelihood function of (2), and the standard M-step (5) finds the maximizer of this lower bound. In general, this maximization can only be solved approximately. For example, the "gradient EM" algorithm considered in [4] performs one gradient ascent step on the $Q(\cdot \mid \boldsymbol{\beta})$ function.

The least-squares M-step (4) may also be viewed as an approximation to the standard M-step (5), as we observe numerically (see Appendix H.3) that the LS-EM update $\boldsymbol{\beta}^+$ satisfies

$$Q(\boldsymbol{\beta}^+ \mid \boldsymbol{\beta}) > Q(\boldsymbol{\beta} \mid \boldsymbol{\beta}) \quad \text{if } \boldsymbol{\beta} \neq \boldsymbol{\beta}^*. \qquad (6)$$

This implies that the least-squares M-step finds an improved solution $\boldsymbol{\beta}^+$ (compared to the previous iterate $\boldsymbol{\beta}$) of the function $Q(\cdot \mid \boldsymbol{\beta})$.

# 4 Analysis of Least Squares EM

In this section, we analyze the convergence behavior of the LS-EM update (4) in the population setting. We first consider the one dimensional case ($d = 1$) in Section 4.1 and establish the global convergence of LS-EM, extending the techniques in [10] for 2GMM to log-concave mixtures. In

Section 4.2, we prove global convergence in the multi-dimensional case ($d > 1$). In this setting, the LS-EM update is *not* contractive in $\ell_2$, so the analysis requires the new ingredient of an angle decreasing property.

For convenience, we introduce the shorthand $F_{\boldsymbol{\beta},\sigma}(\boldsymbol{X}) := g\left(\frac{1}{\sigma}\|\boldsymbol{X} + \boldsymbol{\beta}\|_2\right) - g\left(\frac{1}{\sigma}\|\boldsymbol{X} - \boldsymbol{\beta}\|_2\right)$; when $\sigma = 1$, we simply write $F_{\boldsymbol{\beta}} \equiv F_{\boldsymbol{\beta},1}$. Since the integrand in (4) is an even function of $\boldsymbol{X}$, the update (4) can be simplified to an equivalent form by integrating over one component of the mixture:

$$\boldsymbol{\beta}^+ = M(\boldsymbol{\beta}^*, \boldsymbol{\beta}) = \mathbb{E}_{\boldsymbol{X} \sim f_{\boldsymbol{\beta}^*,\sigma}} \boldsymbol{X} \tanh\left(0.5 F_{\boldsymbol{\beta},\sigma}(\boldsymbol{X})\right). \tag{7}$$

Throughout the section, we refer to the technical conditions permitting the interchange of differentiation and integration as the *regularity condition*. This condition is usually satisfied by log-concave distributions — a detailed discussion is provided in Appendix E.

## 4.1 One Dimensional Case ($d = 1$)

For one dimensional log-concave mixtures, the behavior of LS-EM is similar to that of EM algorithm for 2GMM: there exist only 3 fixed points, $0$, $\beta^*$ and $-\beta^*$, among which $0$ is non-attractive. Consequently, LS-EM converges to the true parameter ($\beta^*$ or $-\beta^*$) from any non-zero initial solution $\beta^0$. This is established in the following theorem.

**Theorem 4.1** (Global Convergence, 1D). *Suppose that $f \in \mathcal{F}$ satisfies the regularity condition. The LS-EM update* (4), $\beta \mapsto M(\beta^*, \beta)$, *has exactly three fixed points:* $0$, $\beta^*$ *and* $-\beta^*$. *Moreover, the following one-step bound holds:*

$$|M(\beta^*, \beta) - \text{sign}(\beta\beta^*)\beta^*| \leq \kappa(\beta^*, \beta, \sigma) \cdot |\beta - \text{sign}(\beta\beta^*)\beta^*|,$$

*where the contraction factor*

$$\kappa(\beta^*, \beta, \sigma) := \mathbb{E}_{X \sim f_{\min(|\beta|,|\beta^*|),\sigma}}\left[1 - \tanh\left(0.5 F_{\min(|\beta|,|\beta^*|),\sigma}(X)\right)\right]$$

*satisfies* $0 < \kappa(\beta^*, \beta, \sigma) < 1$ *when* $\beta \notin \{0, \beta^*, -\beta^*\}$.

We prove this theorem in Appendix C.1. The crucial property used in the proof is the self-consistency of the LS-EM update (4), namely $M(\beta, \beta) = \beta$ for all $\beta$. This property allows us to extend the sensitivity analysis technique for 2GMM to general log-concave distributions.

It can be further shown that the contraction factor $\kappa(\beta^*, \beta, \sigma)$ becomes smaller as the iterate moves closer to the true $\boldsymbol{\beta}^*$ (see Lemma C.2). We thus obtain the following corollary on global convergence at a geometric rate. Without loss of generality, assume $\beta^* > 0$.

**Corollary 4.2** ($t$-step Convergence Rate, 1D). *Suppose that $f \in \mathcal{F}$ satisfies the regularity condition. Let $\beta^t$ denote the output of LS-EM after $t$ iterations, starting from $\beta^0 \neq 0$. The following holds:*

$$|\beta^t - \text{sign}(\beta^0\beta^*)\beta^*| \leq \kappa(\beta^*, \beta^0, \sigma)^t \cdot |\beta^0 - \text{sign}(\beta^0\beta^*)\beta^*|.$$

*If $\beta^0$ is in $(0, 0.5\beta^*)$ or $(1.5\beta^*, \infty)$, running LS-EM for $O\left(\log\frac{0.5\beta^*}{|\beta^0 - \beta^*|}/\log\kappa(\beta^*, \beta^0, \sigma)\right)$ iterations outputs a solution in $(0.5\beta^*, 1.5\beta^*)$. In addition, if $\beta^0$ is in $(0.5\beta^*, 1.5\beta^*)$, running LS-EM for $O\left(C_f(\eta)\log(1/\epsilon)\right)$ iterations outputs an $\epsilon$-close estimate of $\beta^*$, where $C_f(\eta) > 0$ is a constant depending only on $f$ and the SNR $\eta := \beta^*/\sigma$.*

**Special cases** We provide explicit convergence rates for mixtures of some common log-concave distributions. In the following, we assume $\beta^* > 0$ and $\beta \geq 0$ without loss of generality, and set $z := \min(\beta, \beta^*)$.

- Gaussian: $\kappa(\beta^*, \beta, \sigma) \leq \exp\left(-z^2/2\sigma^2\right)$ and $C_f(\eta) = \max\left(1, \frac{1}{\eta^2}\right)$.

- Laplace: $\kappa(\beta^*, \beta, \sigma) \leq \frac{2\exp(-\frac{\sqrt{2}}{\sigma}z)}{1+\exp(-2\frac{\sqrt{2}}{\sigma}z)}$ and $C_f(\eta) = \max\left(1, \frac{1}{\eta}\right)$.

- Logistic: $\kappa(\beta^*, \beta, \sigma) \leq \frac{4\exp(-\frac{\pi z}{\sigma\sqrt{3}})}{1+\exp(-\frac{2\pi z}{\sigma\sqrt{3}})+2\exp(-\frac{\pi z}{\sigma\sqrt{3}})}$ and $C_f(\eta) = \max\left(1, \frac{1}{\eta}\right)$.

See Appendix C.2 for the proofs of the above results. Note that the convergence rate depends on the signal-to-noise ratio $\eta$ as well as the asymptotic growth rate $\gamma \equiv \gamma_f$ of the log-density function $g = -\log f$. In the above examples, $\kappa(\beta^*, \beta, \sigma) \approx \exp\left(-c(\min(\beta^*, \beta)/\sigma)^\gamma\right)$, where $\gamma = 1$ for Laplace and Logistic, and $\gamma = 2$ for Gaussian.

## 4.2 High Dimensional Case ($d > 1$)

Extension to higher dimensions is more challenging for log-concave mixtures than for Gaussian mixtures. Unlike Gaussian, a log-concave distribution with diagonal covariance need not have independent coordinates. A more severe challenge arises due to the fact that *LS-EM is not contractive in $\ell_2$ distance* for general log-concave mixtures. This phenomenon, proved in the lemma below, stands in sharp contrast to the Gaussian mixture setting.

**Lemma 4.3** (Non-contraction in $\ell_2$). *Consider a log-concave density of the form $g(\boldsymbol{x}) \propto \|\boldsymbol{x}\|_2^r$ with $r \geq 1$. When $r \in [1, 2]$, $\boldsymbol{0}$ is the only fixed point of LS-EM in the direction ortoghonal to $\boldsymbol{\beta}^*$. When $r \in (2, \infty)$, there exists a fixed point other than $\boldsymbol{0}$ in the orthogonal direction. Consequently, when $r > 2$, there exists $\boldsymbol{\beta}$ such that $\|M(\boldsymbol{\beta}^*, \boldsymbol{\beta}) - \boldsymbol{\beta}^*\|_2 > \|\boldsymbol{\beta} - \boldsymbol{\beta}^*\|_2$.*

We prove Lemma 4.3 in Appendix D.3. The lemma shows that it is fundamentally impossible to prove global convergence of LS-EM solely based on $\ell_2$ distance, which was the approach taken in [10] for Gaussian mixtures.

Despite the above challenges, we show *affirmatively* that LS-EM converges globally to $\pm\boldsymbol{\beta}^*$ for mixtures of rotation-invariant log-concave distributions, as long as the initial iterate is not orthogonal to $\boldsymbol{\beta}^*$ (a measure zero set).

As the first step, we use rotation invariance to show that the LS-EM iterates stay in a two-dimensional space. The is done in the following lemma, which is proved in Appendix D.1.

**Lemma 4.4** (LS-EM is 2-Dimensional). *The LS-EM update satisfies: $M(\boldsymbol{\beta}^*, \boldsymbol{\beta}) \in span(\boldsymbol{\beta}, \boldsymbol{\beta}^*)$. Moreover, if $\angle(\boldsymbol{\beta}, \boldsymbol{\beta}^*) = 0$ or $\angle(\boldsymbol{\beta}, \boldsymbol{\beta}^*) = \pi/2$, then $M(\boldsymbol{\beta}^*, \boldsymbol{\beta}) \in span(\boldsymbol{\beta})$.*

We now establish the asymptotic global convergence property of LS-EM.

**Theorem 4.5** (Global Convergence, $d$-Dimensional). *Suppose that $f \in \mathcal{F}$ satisfies the regularity condition. The LS-EM algorithm converges to $sign(\langle \boldsymbol{\beta}^0, \boldsymbol{\beta}^* \rangle)\boldsymbol{\beta}^*$ from any randomly initialized point $\boldsymbol{\beta}^0$ that is not orthogonal to $\boldsymbol{\beta}^*$.*

The theorem is proved using a novel sensitivity analysis that shows decrease *in angle* rather than in $\ell_2$ distance. The proof does not depend on the explicit form of the density, but only log-concavity and rotation invariance. We sketch the main ideas of proof below, deferring the details to Appendix D.2.

*Proof Sketch.* Let $\boldsymbol{\beta}^0$ be the initial point that is not orthogonal to $\boldsymbol{\beta}^*$. Without loss of generality, we assume $\langle \boldsymbol{\beta}^0, \boldsymbol{\beta}^* \rangle > 0$. Consequently, all the future iterates satisfy $\langle \boldsymbol{\beta}^t, \boldsymbol{\beta}^* \rangle > 0$ (see Lemma D.4).

If $\boldsymbol{\beta}^0$ is in the span of $\boldsymbol{\beta}^*$ (i.e., $\boldsymbol{\beta}^0$ parallels $\boldsymbol{\beta}^*$), Lemma D.3 ensures that the iterates remain in the direction of $\boldsymbol{\beta}^*$ and converge to $\boldsymbol{\beta}^*$. On the other hand, if $\boldsymbol{\beta}^0$ is not in the span of $\boldsymbol{\beta}^*$, we make use of the following two key properties of the LS-EM update $\boldsymbol{\beta}^+ = M(\boldsymbol{\beta}^*, \boldsymbol{\beta})$:

1. *Angle Decreasing Property* (Lemma D.2): Whenever $\angle\boldsymbol{\beta}, \boldsymbol{\beta}^* \in (0, \frac{\pi}{2})$, the LS-EM update strictly decreases the iterate's angle toward $\boldsymbol{\beta}^*$, i.e., $\angle(\boldsymbol{\beta}^+, \boldsymbol{\beta}^*) < \angle(\boldsymbol{\beta}, \boldsymbol{\beta}^*)$ ;
2. *Local Contraction Region* (Corollary D.7): there is a local region around $\boldsymbol{\beta}^*$ such that if any iterate falls in that region, all the future iterates remain in that region.

Since the sequence of LS-EM iterates is bounded, it must have accumulation points. Using the angle decreasing property and the continuity of $M(\boldsymbol{\beta}^*, \boldsymbol{\beta})$ in the second variable $\boldsymbol{\beta}$, we show that all the accumulation points must be in the direction of $\boldsymbol{\beta}^*$. In view of the dynamics of the 1-dimensional case (Theorem 4.1), we can further show that the set of accumulation points must fall into one of the following three possibilities: $\{\boldsymbol{0}\}$, $\{\boldsymbol{\beta}^*\}$, or $\{\boldsymbol{0}, \boldsymbol{\beta}^*\}$. Below we argue that $\{\boldsymbol{0}\}$ and $\{\boldsymbol{0}, \boldsymbol{\beta}^*\}$ are impossible by contradiction.

- If $\{\boldsymbol{0}\}$ is the set of accumulation points, the sequence of non-zero iterates $\{\boldsymbol{\beta}^t\}$ would converge to $\boldsymbol{0}$ and stay in a neighborhood of $\boldsymbol{0}$ after some time $T$; in this case, Lemma D.8 states that the norm of the iterates is bounded away from zero in the limit and hence they cannot converge to $\boldsymbol{0}$.
- If $\{\boldsymbol{0}, \boldsymbol{\beta}^*\}$ is the set of accumulation points, then there is at least one iterate in the local region of $\boldsymbol{\beta}^*$; by the local contraction region property above, all the future iterates remain close to $\boldsymbol{\beta}^*$. Therefore, $\boldsymbol{0}$ cannot be another accumulation point.

We conclude that $\boldsymbol{\beta}^*$ is the only accumlation point, which LS-EM must converge to. $\square$

# 5 Finite Sample Analysis

In this section, we consider the finite sample setting in which we are given $n$ data points $\boldsymbol{X}^i$ sampled i.i.d. from $D(\boldsymbol{\beta}^*, \sigma)$. Using the equivalent expression (7) for the population LS-EM update, and replacing the expectation with the sample average, we obtain the finite-sample LS-EM update:[3]

$$\widetilde{\boldsymbol{\beta}}^+ = \frac{1}{n} \sum_{i=1}^n \boldsymbol{X}^i \tanh(0.5 F_{\boldsymbol{\beta},\sigma}(\boldsymbol{X}^i)), \qquad \text{where } \boldsymbol{X}^i \overset{\text{i.i.d.}}{\sim} f_{\boldsymbol{\beta}^*,\sigma}. \tag{8}$$

One approach to extend the population results (in Section 4) to this case is by coupling the population update $\boldsymbol{\beta}^+$ with the finite-sample update $\widetilde{\boldsymbol{\beta}}^+$. To this end, we make use of the fact that log-concave distributions are automatically sub-exponential (see Lemma F.2), so the random variables $\{X_j^i \tanh(0.5 F_{\boldsymbol{\beta},\sigma}(\boldsymbol{X}^i))\}_{i=1}^n$ are i.i.d. sub-exponential for each coordinate $j$. Therefore, the concentration bound $\|\widetilde{\boldsymbol{\beta}}^+ - \boldsymbol{\beta}^+\|_2 = \widetilde{O}\big(\sqrt{(\|\boldsymbol{\beta}^*\|_2^2 + \sigma^2)d/n}\big)$ holds, and we expect that the convergence properties of the population LS-EM carry over to the finite-sample case, modulo a statistical error of $\widetilde{O}\big(\sqrt{d/n}\big)$.

The above argument is made precise in following proposition for the one-dimensional case, which is proved in Appendix F.1.

**Proposition 5.1** (1-d Finite Sample). *Suppose the density function $f \in \mathcal{F}$ satisfies the regularity condition. With $\beta \in \mathbb{R}$ being the current estimate, the finite-sample LS-EM update* (8) *satisfies the following bound with probability at least $1 - \delta$:*

$$|\widetilde{\beta}^+ - \beta^*| \le \kappa(\beta^*, \beta, \sigma) \cdot |\beta - \beta^*| + (\beta^* + C_f \sigma) \cdot O\left(\sqrt{\frac{1}{n}\log\frac{1}{\delta}}\right), \tag{9}$$

*where $\kappa(\beta^*, \beta, \sigma)$ is contraction factor defined in Theorem 4.1 and $C_f$ is the Orlicz $\Psi_1$ norm (i.e., the sub-exponential parameter) of a random variable with density $f \in \mathcal{F}$.*

Using Proposition 5.1, we further deduce the global convergence of LS-EM in the finite sample case, which parallels the population result in Corollary 4.2. We present this result assuming sample splitting, i.e., each iteration uses a fresh, independent set of samples. This assumption is standard in finite-sample analysis of EM [4, 29, 10, 28, 17, 16]. In this setting, we establish the following quantitative convergence guarantee for LS-EM initialized at any non-zero $\beta^0$. Without loss of generality, $\beta^0, \beta^* > 0$.

The convergence has two stages. In the first stage, the LS-EM iterates enter a local neighborhood around $\beta^*$, regardless of whether $\beta^0$ is close to or far from 0. This is the content of the result below.

**Proposition 5.2** (First Stage: Escape from 0 and $\infty$). *Suppose the initial point $\beta^0$ is in $(0, 0.5\beta^*) \cup (1.5\beta^*, \infty)$. After $T = O\left(\log\frac{0.25\beta^*}{|\beta^0 - \beta^*|}/\log\kappa(\beta^*, \min(\beta^0, 0.5\beta^*), \sigma)\right)$ iterations, with $N/T = \Omega\left(\frac{(1 + C_f/\eta)^2}{(1 - \kappa(\beta^*, \min(\beta^0, 0.5\beta^*), \sigma))^2}\log\frac{1}{\delta}\right)$ fresh samples per iteration, LS-EM outputs a solution $\widetilde{\beta}^T \in (0.5\beta^*, 1.5\beta^*)$ with probability at least $1 - \delta \cdot O\left(\log\frac{0.25\beta^*}{|\beta^0 - \beta^*|}/\log\kappa(\beta^*, \min(\beta^0, 0.5\beta^*), \sigma)\right)$.*

Within this local neighborhood, the LS-EM iterates converge to $\beta^*$ geometrically, up to a statistical error determined by the sample size. This second stage convergence result is given below.

**Proposition 5.3** (Second Stage: Local Convergence). *The following holds for any $\epsilon > 0$. Suppose $\beta^0 \in (0.5\beta^*, 1.5\beta^*)$. After $T = O(\log\epsilon/\log\kappa(\beta^*, 0.5\beta^*, \sigma))$ iterations, with $N/T = \Omega(\frac{(\beta^* + C_f/\eta)^2}{\epsilon^2(1 - \kappa(\beta^*, 0.5\beta^*, \sigma))^2}\log\frac{1}{\delta})$ fresh samples per iteration, LS-EM outputs a solution $\widetilde{\beta}^T$ satisfying $|\widetilde{\beta}^T - \beta^*| \le \epsilon\beta^*$ with probability at least $1 - \delta \cdot O(\log\epsilon/\log\kappa(\beta^*, 0.5\beta^*, \sigma))$.*

We prove Propositions 5.2 and 5.3 in Appendix F.2.

Let us parse the above results in the special cases of Gaussian, Laplace and Logistic, assuming that $\sigma = 1$ for simplicity. In Section 4.1 we showed that $\kappa(\beta^*, \beta, \sigma) = \exp\big(-\min(\beta, \beta^*)^\gamma\big)$,

where $\gamma \equiv \gamma_f$ is the growth rate of the log density $-\log f$. Consequently, the first stage requires $O\left(1/(\min(\beta^0, \beta^*))^\gamma\right)$ iterations with $\widetilde{\Omega}\left(1/(\min(\beta^0, \beta^*))^{2\gamma}\right)$ samples per iteration, and the second stage requires $O\left(\log(1/\epsilon)/\eta^\gamma\right)$ iterations with $\widetilde{\Omega}\left(1/\epsilon^2\eta^{2\gamma}\right)$ samples per iteration, where $\eta := \beta^*/\sigma$ is the SNR. As can be seen, we have better iteration and sample complexities with a larger $\eta \geq 1$ (larger separation between the components) and a larger $\gamma$ (lighter tail of the components).

In contrast, in the low SNR regime with $\eta < 1$, the sample complexity actually becomes worse for a larger $\gamma$ (lighter tails). Indeed, low SNR means that two components are close in location when $\sigma = 1$. If their tails are lighter, then it becomes more likely that the mixture density $(f_{\beta^*,\sigma} + f_{-\beta^*,\sigma})/2$ has a unique mode at 0 instead of two modes at $\pm\beta^*$. In this case, the mixture problem becomes harder as it is more difficult to distinguish between the two components.

In the higher dimensional setting, we can similarly show coupling in $\ell_2$ (i.e., bounding $\|\widetilde{\boldsymbol{\beta}}^+ - \boldsymbol{\beta}^+\|_2$) via sub-exponential concentration. However, extending the convergence results above to $d > 1$ is more subtle, due to the issue of $\ell_2$ non-contraction (see Lemma 4.3). Addressing this issue would require coupling in a different metric (e.g., in angle—see [17, 28]); we leave this to future work.

## 6 Robustness Under Model Mis-specification

In practice, it is sometimes difficult to know a priori the exact parametric form of a log-concave distribution that generates the data. This motivates us to consider the following scenario: the data is from the mixture $D(\boldsymbol{\beta}^*, \sigma)$ in (2) with a true log-concave distribution $f \in \mathcal{F}$ and unknown location parameter $\boldsymbol{\beta}^*$, but we run LS-EM assuming some other log-concave distribution $\widehat{f}(\cdot) = C_{\widehat{g}}^{-1} \exp(-\widehat{g}(\|\cdot\|_2)) \in \mathcal{F}$. Using the same symmetry argument as in deriving (7), we obtain the following expression for the mis-specified LS-EM update in the population case:

$$\widehat{\boldsymbol{\beta}}^+ = \widehat{M}(\boldsymbol{\beta}^*, \boldsymbol{\beta}) := \mathbb{E}_{\boldsymbol{X} \sim f_{\boldsymbol{\beta}^*,\sigma}} \boldsymbol{X} \tanh\left(0.5\widehat{F}_{\boldsymbol{\beta},\sigma}(\boldsymbol{X})\right), \tag{10}$$

where $\widehat{F}_{\boldsymbol{\beta},\sigma}(\boldsymbol{X}) := \widehat{g}\left(\frac{1}{\sigma}\|\boldsymbol{X} + \boldsymbol{\beta}\|_2\right) - \widehat{g}\left(\frac{1}{\sigma}\|\boldsymbol{X} - \boldsymbol{\beta}\|_2\right)$.

Many properties of the LS-EM update are preserved in this mis-specification setting. In particular, using the same approach as in Lemma4.4 and Lemma D.2, we can show that the mis-specified LS-EM update is also a two dimensional object and satisfies the same strict angle decreasing property $\angle(\widehat{\boldsymbol{\beta}}^+, \boldsymbol{\beta}^*) < \angle(\boldsymbol{\beta}, \boldsymbol{\beta}^*)$. Therefore, to study the convergence behavior of mis-specified LS-EM, it suffices to understand the one-dimensional case (i.e., along the $\boldsymbol{\beta}^*$ direction).

We provide such a result focusing on the setting in which $\widehat{f}$ is Gaussian, that is, we fit a Gaussian mixture to a true mixture of log concave distributions. In this setting, we can show that mis-specified LS-EM has only 3 fixed points $\{\pm\overline{\beta}, 0\}$ (Lemma G.1). Moreover, we can bound the distance between $\overline{\beta}$ and the true $\beta^*$, thereby establishing the following convergence result:

**Proposition 6.1** (Fit with 2GMM). *Under the above one dimensional setting with Gaussian $\widehat{f}$, the following holds for some absolute constant $C_0 > 0$: If $\eta \geq C_0$, then the LS-EM algorithm with a non-zero initialization point $\beta^0$ converges to a solution $\overline{\beta}$ satisfying $\mathrm{sign}(\overline{\beta}) = \mathrm{sign}(\beta^0)$ and*

$$\left|\overline{\beta} - \mathrm{sign}(\beta^0\beta^*)\beta^*\right| \leq 10\sigma.$$

We prove this proposition in Appendix G.1. The proposition establishes a robust property of LS-EM: even in the mis-specified setting, LS-EM still converges globally. Moreover, when the SNR $\eta$ is high (i.e., small noise level $\sigma$), the final estimation error is small and scales linearly with $\sigma$.

## 7 Discussion

In this paper, we have established the global convergence of the Least Squares EM algorithm for a mixture of two log-concave densities. Our theoretical results are proved under the following three assumptions on the densities: (i) rotation invariance, (ii) log-concavity, and (iii) monotone increasing of the log density $g$ with respect to the norm; cf. Equation (1). As we discuss in greater details in Appendix H, all these assumptions appear to be essential under our current framework of analysis.

Moreover, empirical results suggest that the log-convexity assumption cannot be relaxed completely: Figure 1 provides an example where the LS-EM algorithm may converge to 0 (an undesired solution) with constant probability when the log-concavity property is violated. See Appendix H for additional numerical results.

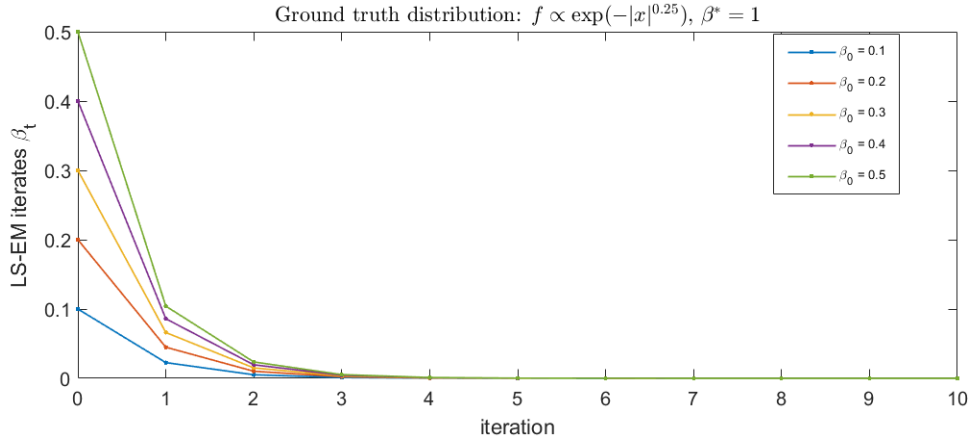

Figure 1: The base distribution class $f$ is proportional to $\exp(-|x|^{0.25})$. This is log-convex instead of log-concave. The ground truth location parameter is set to be 1. We initialize the LS-EM iterates at $\beta = 0.1, 0.2, 0.3, 0.4, 0.5$, and it is seen that all converge to 0 after some iterations.

For future work, an immediate direction is to establish quantitative global convergence guarantees in high dimensions for both the population and finite sample cases; doing so would require generalizing the angle convergence property in [17] to log-concave distributions. In light of the discussion above, it is also of interest to investigate to what extent the rotation invariance assumption can be relaxed, as many interesting log-concave distributions are skewed. It is also interesting to consider mixtures with multiple components.

**Acknowledgement**

W. Qian and Y. Chen are partially supported by NSF grants CCF-1657420 and CCF-1704828.

## Footnotes

[1]Familiar examples of log-concave distributions include Gaussian, Laplace, Gamma, and Logistics [3].

[2]Note that $\boldsymbol{x} \mapsto g(\|\boldsymbol{x}\|_2)$ is a convex function, as it is the composition of a convex function and a convex increasing function. The normalization constant $C_g$ can be computed explicitly by $C_g = C_h d v_d$ with $C_h = \int_0^\infty t^{d-1} \exp(-g(t)) \, dt$, where $v_d := \frac{\pi^{d/2}}{\Gamma(d/2+1)}$ is the volume of a unit ball in $\mathbb{R}^d$.

[3]This expression is for analytic purpose only. To actually implement LS-EM, we use samples $\boldsymbol{X}^i$ from the mixture distribution $D(\boldsymbol{\beta}^*, \sigma)$, which is equivalent to (8).
