[Supplementary Material · Global_Convergence_of_for_Demixing_Two_Log_Concave_Densities_appendix.pdf]

# References

[1] Dimitris Achlioptas and Frank McSherry. On spectral learning of mixtures of distributions. In *International Conference on Computational Learning Theory*, pages 458–469. Springer, 2005. 3

[2] Mark Yuying An. Log-concave probability distributions: Theory and statistical testing. *Duke University Dept of Economics Working Paper*, (95-03), 1997. 1, 3

[3] Mark Bagnoli and Ted Bergstrom. Log-concave probability and its applications. *Economic theory*, 26(2):445–469, 2005. 1, 3

[4] Sivaraman Balakrishnan, Martin J. Wainwright, and Bin Yu. Statistical guarantees for the EM algorithm: From population to sample-based analysis. *The Annals of Statistics*, 45(1):77–120, 2017. 1, 3, 4, 7

[5] Babak Barazandeh and Meisam Razaviyayn. On the behavior of the expectation-maximization algorithm for mixture models. In *2018 IEEE Global Conference on Signal and Information Processing (GlobalSIP)*, pages 61–65. IEEE, 2018. 1, 3

[6] Richard E. Barlow and Frank Proschan. Statistical theory of reliability and life testing: probability models. Technical report, Florida State Univ Tallahassee, 1975. 1, 3

[7] Patrick Billingsley. *Probability and measure*. John Wiley & Sons, 2008. 25

[8] Arun Tejasvi Chaganty and Percy Liang. Spectral experts for estimating mixtures of linear regressions. In *International Conference on Machine Learning*, pages 1040–1048, 2013. 3

[9] Madeleine Cule and Richard Samworth. Theoretical properties of the log-concave maximum likelihood estimator of a multidimensional density. *Electronic Journal of Statistics*, 4:254–270, 2010. 3

[10] Constantinos Daskalakis, Christos Tzamos, and Manolis Zampetakis. Ten steps of EM suffice for mixtures of two Gaussians. *arXiv preprint arXiv:1609.00368*, 2016. 1, 2, 3, 4, 6, 7, 34

[11] Arthur P. Dempster, Nan M. Laird, and Donald B. Rubin. Maximum likelihood from incomplete data via the EM algorithm. *Journal of the Royal Statistical Society: Series B (Methodological)*, 39(1):1–22, 1977. 1

[12] Walter R. Gilks and Pascal Wild. Adaptive rejection sampling for gibbs sampling. *Journal of the Royal Statistical Society: Series C (Applied Statistics)*, 41(2):337–348, 1992. 1, 3

[13] Daniel Hsu and Sham M. Kakade. Learning mixtures of spherical Gaussians: moment methods and spectral decompositions. In *Proceedings of the 4th conference on Innovations in Theoretical Computer Science*, pages 11–20. ACM, 2013. 3

[14] Chi Jin, Yuchen Zhang, Sivaraman Balakrishnan, Martin J. Wainwright, and Michael I. Jordan. Local maxima in the likelihood of Gaussian mixture models: Structural results and algorithmic consequences. In *Advances in neural information processing systems*, pages 4116–4124, 2016. 3

[15] Geurt Jongbloed. The iterative convex minorant algorithm for nonparametric estimation. *Journal of Computational and Graphical Statistics*, 7(3):310–321, 1998. 3

[16] Jason M. Klusowski, Dana Yang, and W. D. Brinda. Estimating the coefficients of a mixture of two linear regressions by expectation maximization. *IEEE Transactions on Information Theory*, 2019. 1, 3, 7

[17] Jeongyeol Kwon, Wei Qian, Constantine Caramanis, Yudong Chen, and Damek Davis. Global Convergence of EM Algorithm for Mixtures of Two Component Linear Regression. *arXiv preprint arXiv:1810.05752*, 2018. 1, 3, 7, 8, 9

[18] Bruce G. Lindsay. Mixture models: theory, geometry and applications. In *NSF-CBMS regional conference series in probability and statistics*, pages i–163. JSTOR, 1995. 1

[19] Sai Ganesh Nagarajan and Ioannis Panageas. On the convergence of EM for truncated mixtures of two Gaussians. *arXiv preprint arXiv:1902.06958*, 2019. 1, 3

[20] Nathan Ross. Fundamentals of Stein's method. *Probability Surveys*, 8:210–293, 2011. 34

[21] Kaspar Rufibach. *Log-concave density estimation and bump hunting for IID observations.* PhD thesis, Verlag nicht ermittelbar, 2006. 3

[22] Arora Sanjeev and Ravi Kannan. Learning mixtures of arbitrary Gaussians. In *Proceedings of the thirty-third annual ACM symposium on Theory of computing*, pages 247–257. ACM, 2001. 3

[23] Adrien Saumard and Jon A. Wellner. Log-concavity and strong log-concavity: a review. *Statistics surveys*, 8:45, 2014. 1, 3, 12

[24] D. Michael Titterington, Adrian F. M. Smith, and Udi E. Makov. *Statistical analysis of finite mixture distributions.* Wiley,, 1985. 1

[25] Roman Vershynin. *High-dimensional probability: An introduction with applications in data science*, volume 47. Cambridge University Press, 2018. 27, 28

[26] Guenther Walther. Detecting the presence of mixing with multiscale maximum likelihood. *Journal of the American Statistical Association*, 97(458):508–513, 2002. 1, 3, 12

[27] CF Jeff Wu. On the convergence properties of the EM algorithm. *The Annals of statistics*, 11(1):95–103, 1983. 3

[28] Ji Xu, Daniel J. Hsu, and Arian Maleki. Global analysis of expectation maximization for mixtures of two Gaussians. In *Advances in Neural Information Processing Systems*, pages 2676–2684, 2016. 1, 3, 7, 8

[29] Bowei Yan, Mingzhang Yin, and Purnamrita Sarkar. Convergence of gradient EM on multi-component mixture of Gaussians. In *Advances in Neural Information Processing Systems*, pages 6956–6966, 2017. 3, 7

## A  Notations for Appendix

We use $x \in \mathbb{R}$ and $\boldsymbol{x} \in \mathbb{R}^d$ to denote scalars and vectors, respectively; $X \in \mathbb{R}$ and $\boldsymbol{X} \in \mathbb{R}^d$ to denote scalar and vector random variables, respectively. The $i$-th coordinate of $\boldsymbol{x}$ (or $\boldsymbol{X}$) is $x_i$ (or $X_i$), and the $j$-th data point is denoted by $\boldsymbol{x}^j$ or $\boldsymbol{X}^j$. The Euclidean norm in $\mathbb{R}^d$ is $\| \cdot \|_2$. For two vectors $\boldsymbol{\alpha}, \boldsymbol{\beta} \in \mathbb{R}^d$, we use $\angle(\boldsymbol{\alpha}, \boldsymbol{\beta}) \in (0, \pi)$ to denote the angle between them and $\langle \boldsymbol{\alpha}, \boldsymbol{\beta} \rangle$ to denote their inner product. We use $\widehat{\boldsymbol{\beta}}$ to denote the unit vector of $\boldsymbol{\beta}$, and $\boldsymbol{\beta}^\perp$ to denote a vector orthogonal to $\boldsymbol{\beta}$. Finally, $\boldsymbol{I}_d$ is the $d$-by-$d$ identity matrix, and $\boldsymbol{e}_i \in \mathbb{R}^d$ is the $i$-th standard basis vector.

## B  Elementary Properties of Log-concave Distributions

A function $f : \mathbb{R}^d \to \mathbb{R}$ is *log-concave* if it satisfies:

$$f(\alpha(\boldsymbol{x}) + (1 - \alpha)\boldsymbol{y}) \geq f(\boldsymbol{x})^\alpha f(\boldsymbol{y})^{1-\alpha},$$

for every $\boldsymbol{x}, \boldsymbol{y} \in \mathbb{R}^d$ and $0 \leq \alpha \leq 1$. Equivalently, $\log f$ is a concave function. We consider log-concave distribution $f$ which further satisfies: $\int_{\mathbb{R}^d} f(\boldsymbol{x})d\boldsymbol{x} = 1$. The following is a classical result for log-concave distributions, which says that the log-concavity property is preserved by marginalization and convolution.

**Theorem B.1.** *All marginals as well as the density function of a log-concave distribution is log-concave. The convolution of two log-concave distributions is again a log-concave distribution.*

The log-concave distribution on $\mathbb{R}$ has the following monotone likelihood ratio property:

**Proposition B.2.** *A density function $f$ on $\mathbb{R}$ is log-concave if an only if the translation family $\{f(\cdot - \theta) : \theta \in \mathbb{R}\}$ has a monotone likelihood ratio: for every $\theta_1 < \theta_2$, the ratio $\frac{p(x-\theta_2)}{p(x-\theta_1)}$ is a monotone nondecreasing function of $x$.*

Furthermore, log-concave distribution has finite moments of all order.

**Lemma B.3.** *For a rotation invariant log-concave density:* $\mathbb{R}^d \to \mathbb{R}$, *all the moments exist.*

*Proof.* It suffices to show that $\int_{\boldsymbol{x}} |x_1|^k f(\boldsymbol{x}) \, \mathrm{d}\boldsymbol{x} < \infty$. By the rotation invariant property, we need to show:

$$\int |x_1|^k \int_{x_2, \ldots x_d} f(\boldsymbol{x}) \, \mathrm{d}x_2 \ldots \mathrm{d}x_d \, \mathrm{d}x_1 < \infty. \tag{11}$$

Note that $f_{2:d}(x_1) := \int_{x_2, \ldots x_d} f(\boldsymbol{x}) \, \mathrm{d}x_2 \ldots \mathrm{d}x_d = \exp(-g_{2:d}(x_1))$ is the marginal distribution, thus log-concave by Theorem B.1. The problem is now further reduced to show that a one-dimensional symmetric log-concave distribution $f(x) = \exp(-g(x))$ has finite moments. By the convexity,

$$g(x) \geq g(x_0) + \partial g(x_0)(x - x_0), \tag{12}$$

for some $x_0 > 0$ and $g'(x_0) > 0$. In particular, we have that shown that there exist $x_0, a, b > 0$ such that $g(x) \geq b + a(x - x_0)$. Therefore,

$$
\begin{aligned}
&\int_x |x|^k \exp(-g(x)) \, \mathrm{d}x \\
=\,&2 \int_{x \geq 0} x^k \exp(-g(x)) \, \mathrm{d}x \\
\leq\,&2 \int_{x \geq 0} x^k \exp(-b - a(x - x_0)) \, \mathrm{d}x \\
=\,&2 \exp(-b + ax_0) \int_{x \geq 0} x^k \exp(-ax) \, \mathrm{d}x < \infty.
\end{aligned}
$$

$\square$

We refer the reader to [23] and [26] for a detailed review for other properties of log-concave distributions.

# C   Analysis for $d = 1$

In this section, we prove the convergence results for $d = 1$. Especially, the proof of Theorem 4.1 is presented in Section C.1, and the convergence rate for some explicit log-concave distribution examples is presented in Section C.2.

## C.1   Proof of Theorem 4.1

Recall the shorthand notation:

$$F_{\beta, \sigma}(x) = g\left(\frac{1}{\sigma}|x + \beta|\right) - g\left(\frac{1}{\sigma}|x - \beta|\right).$$

When $\sigma = 1$, we abbreviate $F_{\beta, \sigma}$ as $F_\beta$. For readability, the theorem is restated here:

**Theorem 4.1** (Global Convergence, 1D). *Suppose that $f \in \mathcal{F}$ satisfies the regularity condition. The LS-EM update* (4), $\beta \mapsto M(\beta^*, \beta)$, *has exactly three fixed points:* $0, \beta^*$ *and* $-\beta^*$. *Moreover, the following one-step bound holds:*

$$|M(\beta^*, \beta) - \operatorname{sign}(\beta\beta^*)\beta^*| \leq \kappa(\beta^*, \beta, \sigma) \cdot \left|\beta - \operatorname{sign}(\beta\beta^*)\beta^*\right|,$$

*where the contraction factor*

$$\kappa(\beta^*, \beta, \sigma) := \mathbb{E}_{X \sim f_{\min(|\beta|, |\beta^*|), \sigma}} \left[1 - \tanh\left(0.5 F_{\min(|\beta|, |\beta^*|), \sigma}(X)\right)\right]$$

*satisfies* $0 < \kappa(\beta^*, \beta, \sigma) < 1$ *when* $\beta \notin \{0, \beta^*, -\beta^*\}$.

*Proof.* Without loss of generality, $\beta^* > 0$. It is easy to verify that $M(\beta^*, 0) = 0$, therefore $0$ is a trivial fixed point. We can also assume $\beta > 0, \beta^* > 0$ (in the case for $\beta < 0$, replace $\beta^*$ with $-\beta^*$ in the proof.) By scaling $\beta^* \to \frac{\beta^*}{\sigma}, \beta \to \frac{\beta}{\sigma}$ and $M(\beta^*, \beta) \to \frac{M(\beta^*, \beta)}{\sigma}$, we can further assume that $\sigma = 1$ in the following analysis.

Now we establish the bound for $|M(\beta^*, \beta) - \beta^*|$. When $\beta > \beta^*$,

$$M(\beta^*, \beta) - \beta^* = M(\beta^*, \beta) - M(\beta, \beta) + \beta - \beta^* \tag{13}$$

$$= (\beta - \beta^*) \left( 1 - \frac{\partial M(z, \beta)}{\partial z} \Big|_{z \in (\beta^*, \beta)} \right) \tag{14}$$

$$\leq \sup_{z \in (\beta^*, \beta)} \left( 1 - \frac{\partial M(z, \beta)}{\partial z} \right) (\beta - \beta^*). \tag{15}$$

In (13), we decompose the difference using the self consistency property of the LS-EM operator (Lemma C.1), i.e, $M(\beta, \beta) = \beta$. This allows us to apply the intermediate value theorem for function $M(\cdot, \beta)$ with respect to the first argument in (14). A similar bound holds for the case $\beta < \beta^*$:

$$\beta^* - M(\beta, \beta^*) \leq \sup_{z \in (\beta, \beta^*)} \left( 1 - \frac{\partial M(z, \beta)}{\partial z} \right) (\beta^* - \beta). \tag{16}$$

Combining inequality (15) and (16), we conclude that: If $\beta\beta^* > 0$,

$$|M(\beta^*, \beta) - \beta^*| \leq \underbrace{\sup_{t \in [0,1]} \left[ 1 - \frac{\partial M(z, \beta)}{\partial z} \Big|_{z = t\beta^* + (1-t)\beta} \right]}_{\kappa(\beta^*, \beta)} |\beta - \beta^*|. \tag{17}$$

It remains to upper bound $1 - \frac{\partial M(z, \beta)}{\partial z}$, where $z$ is between $\beta$ and $\beta^*$. The exact expression for $\frac{\partial M(z, \beta)}{\partial z}$ can be computed as follows:

$$M(z, \beta) = \mathbb{E}_{X \sim f_z} X \tanh\left(0.5 F_\beta(X)\right)$$

$$= \int_x f(x - z)(x) \tanh\left(0.5 F_\beta(x)\right) \mathrm{d}x$$

$$= \int_x \underbrace{f(x)(x + z) \tanh\left(0.5 F_\beta(x + z)\right)}_{h(x, z)} \mathrm{d}x. \quad \text{(Change of Variable)}$$

Using the regularity condition, we can differentiate $M(z, \beta)$ with respect to $z$ by interchanging the order of differentiation and integral: $\frac{\partial}{\partial z} \int_x h(x, z) dx = \int_x \frac{\partial}{\partial z} h(x, z) dx$. Note that the expression for $\frac{\partial h(x, z)}{\partial z}$ can be computed easily using the product rule:

$$\frac{\partial h(x, z)}{\partial z} = f(x) \Bigg( \tanh\left(0.5 F_\beta(x + z)\right)$$

$$+ 0.5(x + z) \left( \frac{\partial}{\partial x} F_\beta(x + z) \right) \tanh'\left(0.5 F_\beta(x + z)\right) \Bigg),$$

where in the last step, we use the fact that $\frac{\partial}{\partial x} F_\beta(x + z) = \frac{\partial}{\partial z} F_\beta(x + z)$. Therefore

$$\frac{\partial M(z, \beta)}{\partial z} = \underbrace{\mathbb{E}_{X \sim f_z} \tanh\left(0.5 F_\beta(X)\right)}_{T_1}$$

$$+ \underbrace{\mathbb{E}_{X \sim f_z} \left[ 0.5 X F_\beta'(X) \tanh'\left(0.5 F_\beta(X)\right) \right]}_{T_2}.$$

In Lemma C.2, we show that $T_1, T_2 \geq 0$. Consequently, we obtain a lower bound for $\frac{\partial M(z, \beta)}{\partial z}$ as follows:

$$\frac{\partial M(z, \beta)}{\partial z} \geq \mathbb{E}_{X \sim f_z} \tanh\left(0.5 F_\beta(X)\right) \tag{18}$$

$$\geq \mathbb{E}_{X \sim f_{\min(\beta, \beta^*)}} \tanh\left(0.5 F_{\min(\beta, \beta^*)}(X)\right). \tag{19}$$

Inequality (19) holds since $\mathbb{E}_{X \sim f_z} \tanh\left(0.5F_\beta(X)\right)$ increases with $z$ and $\beta$, which is established in Lemma C.2. Combining inequalities (17) and (19) yields the desired bound:

$$|M(\beta^*, \beta) - \beta^*| \leq \mathbb{E}_{X \sim f_{\min(\beta, \beta^*)}}[1 - \tanh(0.5F_{\min(\beta, \beta^*)}(X))]|\beta - \beta^*|. \tag{20}$$

$\kappa(\beta^*, \beta, \sigma) \in (0, 1)$ follows from Corollary C.5. From (20), we observe that $M(\beta^*, \beta, \sigma)$ moves closer to $\beta^*$ whenever $\beta > 0$ and $\beta \neq \beta^*$, therefore, $\beta^*$ is the unique fixed point on $\beta > 0$. Similarly, $-\beta^*$ is the unique fixed point on $\beta < 0$. We have completed the proof of Theorem 4.1. $\qquad\square$

### C.1.1 Supporting Lemmas for Theorem 4.1

**Lemma C.1.** *The LS-EM operator is consistent:* $M(\beta, \beta) = \beta$ *for all* $\beta$.

*Proof.* This follows from the following algebra:

$$\begin{aligned}
M(\beta, \beta) &= \int_x \frac{1}{2}(f(x - \beta) + f(x + \beta))x \left[\frac{f(x - \beta) - f(x + \beta)}{f(x - \beta) + f(x + \beta)}\right] \mathrm{d}x \\
&= \frac{1}{2} \int_x x(f(x - \beta) - f(x + \beta)) \, \mathrm{d}x \\
&= \frac{1}{2} \int_x (x - \beta)f(x - \beta) \, \mathrm{d}x - \frac{1}{2} \int (x + \beta)f(x + \beta) \, \mathrm{d}x + \beta \\
&= \beta,
\end{aligned}$$

where the last step holds since $x \to xf(x)$ is an odd function. Consequently, the integral $\int_x (x - \beta)f(x - \beta) \, \mathrm{d}x$ and $\int_x (x + \beta)f(x + \beta) \, \mathrm{d}x$ vanishes. $\qquad\square$

**Lemma C.2.** *Suppose the density function* $f$ *satisfies the regularity condition,*

$$\begin{aligned}
\frac{\partial M(z, \beta)}{\partial z} &= \underbrace{\mathbb{E}_{X \sim f_z} \tanh\left(0.5F_\beta(X)\right)}_{T_1} \\
&\quad + \underbrace{\mathbb{E}_{X \sim f_z}\left[0.5XF_\beta'(X)\tanh'\left(0.5F_\beta(X)\right)\right]}_{T_2}.
\end{aligned}$$

*Moreover,* $T_1$ *and* $T_2$ *have following properties:*

- *If* $\beta z > 0$, *then* $T_1 > 0$ *and* $T_2 > 0$;
- $T_1$ *is an increasing function with respect to both* $\beta$ *and* $z$.

*Proof.* Without loss of generality, we assume $z > 0$ and $\beta > 0$. Note that the integrand for both $T_1$ and $T_2$ are odd in $x$. Moreover, they are both strictly positive on a subset of $\{x : x \geq 0\}$ with a positive measure when $\beta > 0$ following the property of $F_\beta(x)$ (See Lemma C.3):

- For $\tanh(0.5F_\beta(x))$, we have that $F_\beta(x) > 0$ on $x \in (0, \infty)$. Thus the integrand for $T_1$ is strictly positive when $x > 0$.
- For $xF_\beta'(x)\tanh'\left(0.5F_\beta(x)\right)$, we have that $x\tanh'\left(0.5F_\beta(x)\right) > 0$ when $x > 0$. Moreover, $F_\beta'(x) > 0$ on a subset of $(0, \infty)$ with positive measure. It follows that $xF_\beta'(x)\tanh'\left(0.5F_\beta(x)\right) > 0$ on a subset of $(0, \infty)$ with positive measure.

Using Lemma C.4, we can show that both $T_1$ and $T_2$ are positive. To see how $T_1$ changes with respect to $\beta$, let us take the derivative with respect to $\beta$ (again, regularity condition allows us to change the order between differentiation and integration):

$$\frac{\partial T_1}{\partial \beta} = \mathbb{E}_{X \sim f_z} 0.5 \frac{\partial F_\beta(X)}{\partial \beta} \tanh'(0.5F_\beta(X)).$$

The integrand is even in $x$. Meanwhile, $\frac{\partial F_\beta(x)}{\partial \beta} \geq 0$ as a function of $x$ and is positive on a subset of $(0, \infty)$ with positive measure from Lemma C.3. Therefore the integrand is again positive on a subset

of $[0, \infty]$ with non-zero measure. We have that $\frac{\partial T_1}{\partial \beta} > 0$ and $T_1$ is an increasing function with respect to $\beta$.

To see how $T_1$ changes with respect to $z$, let us take the derivative with respect to $z$:

$$\frac{\partial T_1}{\partial z} = \mathbb{E}_{X \sim f_z} 0.5 \frac{\partial F_\beta(X)}{\partial X} \tanh'(0.5 F_\beta(X)).$$

The integrand is even in $x$. $\frac{\partial F_\beta(X)}{\partial X} \geq 0$ and is positive on a subset of $(0, \infty)$ with positive measure from Lemma C.3. Therefore, $\frac{\partial T_1}{\partial z} > 0$ and $T_1$ is increasing with respect to $z$. $\qquad \square$

**Lemma C.3** (Property of $F_\beta$). *$F_\beta(x)$ is an odd function in $x$ with $F_\beta(x) > 0$ on $(0, \infty)$. Both $\frac{\partial F_\beta(x)}{\partial \beta}$ and $\frac{\partial F_\beta(x)}{\partial x}$ are non-negative on $[0, \infty]$. Moreover, they are strictly positive as a function of $x$ on a subset of $[0, \infty]$.*

*Proof.* $F_\beta(x)$ is an odd function in $x$ since

$$
\begin{aligned}
F_\beta(-x) &= g(|-x+\beta|) - g(|-x-\beta|) \\
&= g(|x-\beta|) - g(|x+\beta|) = -F_\beta(x).
\end{aligned}
$$

It follows that $\frac{\partial F_\beta(x)}{\partial x}$ and $\frac{\partial F_\beta(x)}{\partial \beta}$ are even functions in $x$. When $x \geq 0$,

$$
F_\beta(x) = \begin{cases} g(x+\beta) - g(\beta - x) & x \in [0, \beta] \\ g(x+\beta) - g(x - \beta) & x \in (\beta, \infty). \end{cases}
$$

Since $g$ is strictly increasing on $(0, \infty)$, it is easy to see that $F_\beta(x) > 0$ when $x > 0$. Taking the partial derivative with respect to $x$ and $\beta$ respectively, we have:

$$
\frac{\partial F_\beta(x)}{\partial \beta} = \begin{cases} g'(x+\beta) - g'(\beta - x) & x \in [0, \beta] \\ g'(x+\beta) + g'(x - \beta) & x \in (\beta, \infty); \end{cases}
$$

and

$$
\frac{\partial F_\beta(x)}{\partial x} = \begin{cases} g'(x+\beta) + g'(\beta - x) & x \in [0, \beta] \\ g'(x+\beta) - g'(x - \beta) & x \in (\beta, \infty). \end{cases}
$$

Since $g$ is convex, the derivative is non-decreasing. It follows that both $g'(x+\beta) - g'(\beta - x) \geq 0$ and $g'(x+\beta) - g'(x-\beta) \geq 0$ in the range $[0, \beta]$, and $(\beta, \infty)$ respectively. Moreover, $g$ is strictly increasing on $[0, \infty)$, it must have positive derivative in $(0, \infty)$. Therefore, $g'(x+\beta) + g'(x-\beta) > 0$ and $g'(x+\beta) + g'(\beta - x) > 0$ on the range $(\beta, \infty)$ and $([0, \beta])$ respectively. $\qquad \square$

**Lemma C.4** (positive integral). *Let $f \in \mathcal{F}$. Let $S \subseteq \mathbb{R}$ be a set with non-zero measure. Suppose that $h$ is an odd function with $h(x) \geq 0$ on $[0, \infty]$ and $h(x) > 0$ on $S$. If $z > 0$, the following holds:*

$$\mathbb{E}_{X \sim f_z} h(X) > 0. \tag{21}$$

*Proof.*

$$
\begin{aligned}
\mathbb{R}_{X \sim f_z} h(X) &= \int_x f(x-z) h(x) \, \mathrm{d}x \\
&= \int_{x \geq 0} f(x-z) h(x) \, \mathrm{d}x + \int_{x \leq 0} f(x-z) h(x) \, \mathrm{d}x \\
&= \int_{x \geq 0} f(x-z) h(x) \, \mathrm{d}x + \int_{x \geq 0} f(x+z) (-h(x)) \, \mathrm{d}x \tag{22} \\
&= \int_{x \geq 0} (f(x-z) - f(x+z)) h(x) \, \mathrm{d}x \\
&= \int_{x \in S} (f(x-z) - f(x+z)) h(x) \, \mathrm{d}x
\end{aligned}
$$

(22) holds since $h$ is an odd function: $h(-x) = -h(x)$. Since $f \in \mathcal{F}$, $f(x-z) - f(x+z) > 0$ on $x \geq 0$. We thus conclude that the above integral is positive. $\qquad \square$

**Corollary C.5.** *If* $z > 0$, $0 < \mathbb{E}_{x \sim z}(1 - \tanh(\frac{g(x+z)-g(x-z)}{2})) < 1$.

*Proof.* Since $|\tanh(\cdot)| < 1$,

$$\mathbb{E}_{X \sim f_z} \tanh(F_z(X)) < 1,$$

the left hand side of the inequality is proved. For the right hand side, we use the previous argument that $(1) F_z(x) > 0$ and $(2) F_z(\cdot)$ is odd to conclude that

$$\mathbb{E}_{X \sim f_z} \tanh(F_z(X)) > 0$$

by Lemma C.4. The right hand side of the inequality is proved. $\square$

### C.2 Convergence Rate for Special Distributions

In the following we compute an explicit convergence rate for popular distributions. Without loss of generality, we can assume $\sigma = 1$ and replace $\beta$ by $\frac{\beta}{\sigma}$ in the formal statement.

- Gaussian: $f(x) = \frac{1}{\sqrt{2\pi}} \exp\left(-\frac{x^2}{2}\right)$, and $\tanh(0.5 F_\beta(x)) = \tanh(\beta x)$. We compute $\kappa(\beta, \beta^*, \sigma)$ as follows: for any $\beta > 0$,

$$\mathbb{E}_{X \sim f_\beta} [1 - \tanh(\beta X)]$$

$$= \mathbb{E}_{X \sim f_\beta} \frac{2 \exp(-\beta X)}{\exp(-\beta X) + \exp(\beta X)}$$

$$\leq \mathbb{E}_{X \sim f_\beta} \exp(-\beta X) = \exp\left(-\frac{\beta^2}{2}\right).$$

**Corollary C.6** (Gaussian). $\kappa(\beta, \beta^*, \sigma) = \exp\left(-\frac{\min(|\beta|,|\beta^*|)^2}{2\sigma^2}\right)$.

- Laplace: $f(x) = \frac{1}{2} \exp(-|x|)$, and $\tanh(0.5 F_\beta(x)) = \frac{\exp(-|x-\beta|) - \exp(-|x+\beta|)}{\exp(-|x-\beta|) + \exp(-|x+\beta|)}$.

$$\mathbb{E}_{X \sim f_\beta} [1 - \tanh(0.5 F_\beta(X))]$$

$$= \int_x \exp(-|x - \beta|) \frac{\exp(-|x + \beta|)}{\exp(-|x - \beta|) + \exp(-|x + \beta|)} \, \mathrm{d}x$$

$$= \int_{x \leq -\beta} \exp(x - \beta) \frac{\exp(x + \beta)}{\exp(x - \beta) + \exp(x + \beta)} \, \mathrm{d}x$$

$$+ \int_{x \geq \beta} \exp(-x + \beta) \frac{\exp(-x - \beta)}{\exp(-x + \beta) + \exp(-x - \beta)} \, \mathrm{d}x$$

$$+ \int_{x \in (-\beta, \beta)} \exp(x - \beta) \frac{\exp(-x - \beta)}{\exp(x - \beta) + \exp(-x - \beta)} \, \mathrm{d}x$$

$$= 2 \frac{\exp(-\beta)}{\exp(-\beta) + \exp(+\beta)} + \exp(-\beta) \int_{x=-\beta}^{\beta} \frac{1}{\exp(x) + \exp(-x)} \, \mathrm{d}x$$

$$\leq 2 \frac{\exp(-\beta)}{\exp(-\beta) + \exp(+\beta)} + 2 \frac{1}{1 + \exp(-2\beta)} \exp(-\beta)(1 - \exp(-\beta)) \qquad (23)$$

$$= 2 \frac{\exp(-\beta)}{1 + \exp(-2\beta)}$$

where in (23) we used the numerical inequality:

$$\frac{1}{\exp(x) + \exp(-x)} \leq \frac{1}{1 + \exp(-2\beta)} \exp(-x) \quad \forall x \in (0, \beta).$$

**Corollary C.7** (Laplace). $\kappa(\beta, \beta^*, \sigma) = \frac{2 \exp(-\frac{\min(|\beta|,|\beta^*|)}{\sigma/\sigma_0})}{1 + \exp(-2 \frac{\min(|\beta|,|\beta^*|)}{\sigma/\sigma_0})}$. *with* $\sigma_0 = \sqrt{2}$ *is the number that makes the Laplace distribution has variance* 1 *for normalization purpose.*

- Logistic: $f(x) = \frac{\exp(x)}{(1+\exp(x))^2}$, and $\tanh(0.5F_\beta(x)) = \frac{(\exp(\beta)-\exp(-\beta))(\exp(2x)-1)}{(\exp(\beta)+\exp(-\beta))(1+\exp(2x))+4\exp(x)}$.

$$\mathbb{E}_{X \sim f_\beta}\left[1 - \tanh(0.5F_\beta(X))\right]$$

$$=2\int \frac{\frac{\exp(x-\beta)}{(1+\exp(x-\beta))^2}\frac{\exp(x+\beta)}{(1+\exp(x+\beta))^2}}{\frac{\exp(x-\beta)}{(1+\exp(x-\beta))^2} + \frac{\exp(x+\beta)}{(1+\exp(x+\beta))^2}}\,\mathrm{d}x$$

$$=2\int \frac{\exp(x)}{\exp(-\beta)(1+\exp(x+\beta))^2 + \exp(\beta)(1+\exp(x-\beta))^2}\,\mathrm{d}x$$

$$=2\int \frac{\exp(x)}{\exp(-\beta)+\exp(\beta)+4\exp(x)+\exp(2x)(\exp(\beta)+\exp(-\beta))}\,\mathrm{d}x$$

$$=2\int \frac{1}{(\exp(-x)+\exp(x))(\exp(-\beta)+\exp(\beta))+4}\,\mathrm{d}x$$

$$=4\int_{x\geq 0} \frac{1}{(1+\exp(2x))(\exp(-\beta)+\exp(\beta))+4\exp(x)}\,d\left[\exp(x)\right]$$

$$=4\int_{s=1}^{\infty} \frac{1}{(1+s^2)c+4s}\,ds,$$

where $c = \exp(-\beta) + \exp(\beta)$. We can further upper bound the last integral by the following:

$$=4\int_{s=1}^{\infty} \frac{1}{(\sqrt{c}s + \frac{2}{\sqrt{c}})^2 + c - \frac{4}{c}}\,ds$$

$$\leq 4\frac{1}{\sqrt{c}}[-\frac{1}{x}]_{x=\sqrt{c}+\frac{2}{\sqrt{c}}}^{\infty}$$

$$=\frac{4}{\sqrt{c}}\frac{1}{\sqrt{c}+\frac{2}{\sqrt{c}}} = \frac{4}{c+2} = \frac{4}{\exp(\beta)+\exp(-\beta)+2} < 1.$$

where in the second step, we use the fact that $c \geq 2$.

**Corollary C.8** (Logistic). $\kappa(\beta, \beta^*, \sigma) = \frac{4}{\exp\left(\frac{\min(|\beta|,|\beta^*|)}{\sigma/\sigma_0}\right)+\exp\left(-\frac{\min(|\beta|,|\beta^*|)}{\sigma/\sigma_0}\right)+2}$ *with* $\sigma_0 = \pi/\sqrt{3}$ *is the number that makes the logistic distribution has variance* $1$ *for normalization purpose.*

# D   Analysis for $d > 1$

In this section, we prove the convergence result for the setting $d > 1$. In Section D.1, we prove for Lemma 4.4 that shows the population LS-EM update is two dimensional; in Section D.2, we present the proof for Theorem 4.5 on the asymptotic convergence to the true location parameter; in Section D.3, we prove Lemma 4.3 that demonstrates the non-contraction phenomenon of the LS-EM update within a general log-concave distribution family.

## D.1   Proof of Lemma 4.4

For readability we restate the lemma below.

**Lemma 4.4** (LS-EM is 2-Dimensional). *The LS-EM update satisfies:* $M(\beta^*, \beta) \in span(\beta, \beta^*)$. *Moreover, if* $\angle(\beta, \beta^*) = 0$ *or* $\angle(\beta, \beta^*) = \pi/2$, *then* $M(\beta^*, \beta) \in span(\beta)$.

*Proof.* Using the rotation invariance property of the log-concave distribution, the class we are interested in, we use the following local orthonormal basis $\{v_1, \ldots, v_d\}$, with $v_1 = \widehat{\beta}$, and $v_2 = \widehat{\beta}^{\perp}$ satisfying $span(v_1, v_2) = span(\beta, \beta^*)$ and $\langle v_2, \beta^* \rangle \geq 0$. Under this basis, $\beta$ and $\beta^*$ have non-zero entries only in the first two coordinates:

$$\beta = (\|\beta\|_2, 0, \ldots, 0), \quad \beta^* = (\underbrace{\langle \beta^*, \widehat{\beta} \rangle}_{\beta_1^*}, \underbrace{\langle \beta^*, \widehat{\beta}^{\perp} \rangle}_{\beta_2^*}, 0, \ldots, 0).$$

Here we use the shorthand notation $\beta_1^* := \langle \boldsymbol{\beta}^*, \widehat{\boldsymbol{\beta}} \rangle = \|\boldsymbol{\beta}^*\|_2 \cos(\angle \boldsymbol{\beta}, \boldsymbol{\beta}^*)$ and $\beta_2^* := \langle \boldsymbol{\beta}^*, \widehat{\boldsymbol{\beta}}^\perp \rangle = \|\boldsymbol{\beta}^*\|_2 \sin(\angle \boldsymbol{\beta}, \boldsymbol{\beta}^*)$. Meanwhile,

$$\boldsymbol{x} - \boldsymbol{\beta} = (x_1 - \|\boldsymbol{\beta}\|_2, x_2, \ldots, x_d), \quad \boldsymbol{x} - \boldsymbol{\beta}^* = (x_1 - \beta_1^*, x_2 - \beta_2^*, x_3, \ldots, x_d).$$

Recall the expression (7) for the least-squares EM update:

$$\boldsymbol{\beta}^+ = \mathbb{E}_{\boldsymbol{X} \sim f_{\boldsymbol{\beta}^*, \sigma}} \boldsymbol{X} \tanh\left(0.5 F_{\boldsymbol{\beta}, \sigma}(\boldsymbol{X})\right),$$

the $j$-th coordinate of $\boldsymbol{\beta}^+$ is:

$$\beta_j^+ = \mathbb{E}_{\boldsymbol{x} \sim f_{\boldsymbol{\beta}^*, \sigma}} x_j \tanh\left(0.5 F_{\boldsymbol{\beta}, \sigma}(\boldsymbol{x})\right) \tag{24}$$

$$= \int_{x_{-j}} \int_{x_j} \underbrace{\frac{1}{\sigma^d C_g} \exp\left(-g\left(\|\frac{1}{\sigma}(\boldsymbol{x} - \boldsymbol{\beta}^*)\|_2\right)\right) x_j \tanh\left(0.5 F_{\boldsymbol{\beta}, \sigma}(\boldsymbol{x})\right) \mathrm{d}x_j}_{h_j} \, \mathrm{d}x_{-j},$$

where $x_{-j}$ denotes all the coordinates that is not $x_j$. It is easy to see $h_j$ is an odd function in $x_j$ when $j \geq 3$, therefore $\beta_j^+ = 0$ for all $j \geq 3$ and the least-squares M-step preserves the 2 dimensional structure. Moreover, when $\beta_1^* = 0$, i.e, $\boldsymbol{\beta}$ is in the orthogonal direction to $\boldsymbol{\beta}^*$, $\beta_2^+ = 0$ as $h_j$ is an odd function in $x_1$. When $\beta_2^* = 0$, i.e, $\boldsymbol{\beta}$ is in the same direction as $\boldsymbol{\beta}^*$, $\beta_2^+ = 0$ as $h_j$ is an odd function in $x_2$. In other words, $\text{Span}(\boldsymbol{\beta}^*)$ and $\text{Span}(\boldsymbol{\beta}^{*\perp})$ are 1-dimensional invariant subspaces. $\qquad \square$

## D.2 Proof of Theorem 4.5

From Lemma 4.4, we use the rotation invariance property to rewrite $F_{\boldsymbol{\beta}, \sigma}(\boldsymbol{x})$ as follows:

$$F_{\boldsymbol{\beta}, \sigma}(\boldsymbol{x}) = g\left(\frac{1}{\sigma} \| (x_1 + \|\boldsymbol{\beta}\|_2, x_2, \ldots, x_d) \|_2\right) - g\left(\frac{1}{\sigma} \| (x_1 - \|\boldsymbol{\beta}\|_2, x_2, \ldots, x_d) \|_2\right).$$

$F_{\boldsymbol{\beta}, \sigma}(\boldsymbol{x})$ is an odd function in $x_1$, and it is an even function in $x_2, \ldots, x_d$. When $\sigma = 1$, we use $F_{\boldsymbol{\beta}}$ for the short hand notation. Similar as before, by scaling: $\boldsymbol{\beta} \to \frac{1}{\sigma}\boldsymbol{\beta}, \boldsymbol{\beta}^* \to \frac{1}{\sigma}\boldsymbol{\beta}^*, M(\boldsymbol{\beta}^*, \boldsymbol{\beta}) \to \frac{1}{\sigma}M(\boldsymbol{\beta}^*, \boldsymbol{\beta})$, we assume $\sigma = 1$ in the following analysis. $F_{\boldsymbol{\beta}}$ has a similar property as in Lemma C.3 in the 1-D case.

**Corollary D.1** (property of $F_{\boldsymbol{\beta}}$). *Suppose that $\boldsymbol{\beta} \neq 0$. $F_{\boldsymbol{\beta}}(\boldsymbol{x}) > 0$ when $x_1 > 0$. Both $\frac{\partial F_{\boldsymbol{\beta}}}{x_1}$ and $\frac{\partial F_{\boldsymbol{\beta}}}{\|\boldsymbol{\beta}\|}$ are non-negative. Moreover, they are strictly positive as a function of $x_1$ on a subset of $(0, \infty)$ with positive measure.*

For readability we restate the theorem below.

**Theorem 4.5** (Global Convergence, $d$-Dimensional). *Suppose that $f \in \mathcal{F}$ satisfies the regularity condition. The LS-EM algorithm converges to $\text{sign}(\langle \boldsymbol{\beta}^0, \boldsymbol{\beta}^* \rangle)\boldsymbol{\beta}^*$ from any randomly initialized point $\boldsymbol{\beta}^0$ that is not orthogonal to $\boldsymbol{\beta}^*$.*

*Proof.* Let $\boldsymbol{\beta}^0$ denote an initial point that is not in the orthogonal direction to $\boldsymbol{\beta}^*$. Without loss of generality, we assume $\langle \boldsymbol{\beta}^0, \boldsymbol{\beta}^* \rangle > 0$. There are two cases for $\boldsymbol{\beta}^0$, either $\boldsymbol{\beta}^0$ is in the span of $\boldsymbol{\beta}^*$ or $\boldsymbol{\beta}^0$ is not in the direction of $\boldsymbol{\beta}^*$. In the previous case, the iterates remain in the direction of $\boldsymbol{\beta}^*$ and converge to $\boldsymbol{\beta}^*$ from Lemma D.3.

In the latter case, we argue that all the accumulation points (existence by the boundedness of the iterate) must be in the direction of $\boldsymbol{\beta}^*$. If there exists some $t > 0$ such that $\angle(\boldsymbol{\beta}^t, \boldsymbol{\beta}^*) = 0$, we are reduced to the previous case where the iterates remain in the direction of $\boldsymbol{\beta}^*$. From now on, we assume that $\angle(\boldsymbol{\beta}^t, \boldsymbol{\beta}^*) > 0$ for all $t \geq 0$.

Lemma D.2 establishes the crucial angle decreasing property of the variant EM update, which says that the angle between the iterates and $\boldsymbol{\beta}^*$ strictly decreases, i.e, $\angle(\boldsymbol{\beta}^{t+1}, \boldsymbol{\beta}^*) < \angle(\boldsymbol{\beta}^t, \boldsymbol{\beta}^*)$. Indeed $\{\angle(\boldsymbol{\beta}^t, \boldsymbol{\beta}^*)\}$ is a monotone decreasing sequence, thus this sequence converges to $\theta^\infty \geq 0$, with $\angle(\boldsymbol{\beta}^t, \boldsymbol{\beta}^*) \geq \theta^\infty$ for all $t$.

If $\theta^\infty = 0$, we are done. Otherwise, let $\{\boldsymbol{\beta}^{n_k}\}$ be a subsequence converging to an accumulation point $\boldsymbol{\beta}^\infty$. We deduce that $\angle(\boldsymbol{\beta}^\infty, \boldsymbol{\beta}^*) = \theta^\infty > 0$ since any subsequence of $\{\angle(\boldsymbol{\beta}^t, \boldsymbol{\beta}^*)\}$ converges to $\theta^\infty$.

By the continuity of variant EM operator, the subsequence $\{M(\boldsymbol{\beta}^{n_k}, \boldsymbol{\beta}^*)\}$ converges to $M(\boldsymbol{\beta}^\infty, \boldsymbol{\beta}^*)$. Note that (1) $\{M(\boldsymbol{\beta}^{n_k}, \boldsymbol{\beta}^*)\} = \{\boldsymbol{\beta}^{n_k+1}\}$ and (2) $\angle(M(\boldsymbol{\beta}^\infty, \boldsymbol{\beta}^*), \boldsymbol{\beta}^*) < \theta^\infty$. Thus, there must be some $k$ such that $\angle(\boldsymbol{\beta}^{n_k+1}, \boldsymbol{\beta}^*)$ is strictly between $\angle(M(\boldsymbol{\beta}^\infty, \boldsymbol{\beta}^*), \boldsymbol{\beta}^*)$ and $\theta^\infty$, contradicting with the previous analysis that $\angle(\boldsymbol{\beta}^t, \boldsymbol{\beta}^*) \geq \theta^\infty$ for all $t > 0$. This completes the argument showing that all the accumulation points must be in the direction of $\boldsymbol{\beta}^*$.

Next we show that $\boldsymbol{\beta}^*$ is the only point of convergence.

Let $\mathcal{F}$ be the set of accumulation points of the iterates $\{\boldsymbol{\beta}^t\}$, which are all in the direction of $\boldsymbol{\beta}^*$. Since $\langle \boldsymbol{\beta}^0, \boldsymbol{\beta}^* \rangle > 0$, all the future iterates have positive correlation with $\boldsymbol{\beta}^*$ by Lemma D.4. In particular, it implies that any limit point has non-negative correlation with $\boldsymbol{\beta}^*$. We first show that if $\mathcal{F}$ contains $\boldsymbol{\beta}^*$, then it can not contain elements other than $\boldsymbol{\beta}^*$. Indeed, since $\boldsymbol{\beta}^*$ is a limit point, for any $\epsilon$, there exists $\boldsymbol{\beta}^k$ such that $\|\boldsymbol{\beta}^k - \boldsymbol{\beta}^*\| < \epsilon$ and $\angle(\boldsymbol{\beta}^k, \boldsymbol{\beta}^*)$ is very small. We know that all the iterates after $\boldsymbol{\beta}^k$ remain in a local region of $\boldsymbol{\beta}^*$ of radius $\epsilon$ from Corollary D.7. Thus any other limit points must be inside this local region. Since it holds for arbitrary $\epsilon$, $\boldsymbol{\beta}^*$ is the only limit point.

Similarly, we can show that $\mathcal{F}$ can not contain any non-zero $\boldsymbol{\beta}^\infty \neq \boldsymbol{\beta}^*$. Otherwise, we can use the continuity of the least-squares EM operator to show that there exists some $k$ such that $\boldsymbol{\beta}^k$ falls into a local neighborhood of $\boldsymbol{\beta}^*$ that does not include $\boldsymbol{\beta}^\infty$. (The reason is as follows: since $\boldsymbol{\beta}^\infty \in \mathcal{F}$, there exists $t > 0$ such that $\boldsymbol{\beta}^t$ that is close to $\boldsymbol{\beta}^\infty$. Applying the least-squares EM operator to $\boldsymbol{\beta}^t$ for finitely many times produces an iterate that is very close to $\boldsymbol{\beta}^*$ by Theorem 4.1.) On the other hand, we know that all the iterates after $\boldsymbol{\beta}^k$ remain in a local region of $\boldsymbol{\beta}^*$ that does not include $\boldsymbol{\beta}^\infty$ once it is inside from Corollary D.7, thus $\boldsymbol{\beta}^\infty$ can not be a limit point of the iterates, a contradiction.

There are only two possibilities left: (a) $\{\mathbf{0}\}$, and (b) $\{\boldsymbol{\beta}^*\}$. We argue that (a) is not possible either. (a) implies that $\lim_t \boldsymbol{\beta}^t \to \mathbf{0}$. In particular, there exist $N$ such that for all $n > N$, $\|\boldsymbol{\beta}^n\|_2 \leq \frac{1}{8}\|\boldsymbol{\beta}^*\|_2$. In Lemma D.8, we show that if all the iterates after $\boldsymbol{\beta}^n$ are non-zero and have norm no greater than $\frac{1}{8}\|\boldsymbol{\beta}^*\|_2$, the norm of the iterates must be lower bounded in the limit. Thus they can not converge to $\mathbf{0}$. (b) is the only possibility and we are done. □

### D.2.1 Supporting Lemmas for Theorem 4.5

Below we record several technical lemmas used in the proof of Theorem 4.5.

**Lemma D.2** (Angle Decreasing). *Suppose that the density function $f$ satisfies the regularity condition. $\beta_2^+ > 0$ whenever $\beta_2^* > 0$ or $\beta_1^* > 0$.*

*Proof.* Assume $\sigma = 1$. Define the following function:

$$\beta_2^+(t) = \int_{\boldsymbol{x}} \frac{1}{C_g} \exp\left(-g\left(\|(x_1 - t\beta_1^*, x_2 - \beta_2^*, x_3, \ldots, x_d)\|_2\right)\right) x_2 \tanh(0.5 F_{\boldsymbol{\beta}}(\boldsymbol{x})) \, d\boldsymbol{x}.$$

$\beta_2^+ = \beta_2^+(1)$. We observe that $\beta_2^+(0) = 0$ since the integrand is an odd function in $x_1$. The mean value theorem tells us:

$$\beta_2^+ = \frac{\partial}{\partial t} \beta_2^+(t) \,|_{t \in (0,1)} \,.$$

Under the regularity condition, we can differentiate inside the integral and obtain the following expression for the derivative of $\beta_2^+(t)$ with respect to $t$:

$$\frac{\partial}{\partial t}\beta_2^+(t)$$

$$=\frac{\partial}{\partial t}\int_{\boldsymbol{x}}\frac{1}{C_g}\exp\left(-g\left(\|\left(x_1,x_2-\beta_2^*,x_3,\ldots,x_d\right)\|_2\right)\right)x_2\tanh(0.5F_{\boldsymbol{\beta}}(\boldsymbol{x}+t\beta_1^*\boldsymbol{e}_1))\,\mathrm{d}\boldsymbol{x}$$

$$=0.5\beta_1^*\int_{\boldsymbol{x}}\frac{1}{C_g}\exp\left(-g\left(\|\left(x_1,x_2-\beta_2^*,x_3,\ldots,x_d\right)\|_2\right)\right)x_2\frac{\partial F_{\boldsymbol{\beta}}}{\partial x_1}\tanh'(0.5F_{\boldsymbol{\beta}}(\boldsymbol{x}+t\beta_1^*\boldsymbol{e}_1))\,\mathrm{d}\boldsymbol{x}$$

$$=0.5\beta_1^*\cdot\int_{x_{\overline{2}}}\int_{x_2\geq0}\frac{1}{C_g}\left(f\left(x_1-t\beta_1^*,x_2-\beta_2^*,x_3,\ldots,x_d\right)-f\left(x_1-t\beta_1^*,x_2+\beta_2^*,x_3,\ldots,x_d\right)\right)\cdot$$

$$x_2\left(\frac{\partial}{\partial x_1}F_{\boldsymbol{\beta}}(\boldsymbol{x})\right)\tanh'(0.5F_{\boldsymbol{\beta}}(\boldsymbol{x}))\,\mathrm{d}x_2\,\mathrm{d}x_{-2}$$

$$=0.5\beta_1^*\cdot\int_{x_{\overline{1,2}}}\int_{x_2\geq0,x_1\geq0}\frac{1}{C_g}(f\left(x_1-t\beta_1^*,x_2-\beta_2^*,x_3,\ldots,x_d\right)-f\left(x_1-t\beta_1^*,x_2+\beta_2^*,x_3,\ldots,x_d\right)$$

$$+f\left(x_1+t\beta_1^*,x_2-\beta_2^*,x_3,\ldots,x_d\right)-f\left(x_1+t\beta_1^*,x_2+\beta_2^*,x_3,\ldots,x_d\right))\cdot$$

$$x_2\left(\frac{\partial}{\partial x_1}F_{\boldsymbol{\beta}}(\boldsymbol{x})\right)\tanh'(0.5F_{\boldsymbol{\beta}}(\boldsymbol{x}))\,\mathrm{d}x_2\,\mathrm{d}x_1\,\mathrm{d}x_{bar1,2}.$$

The last step holds as $x_2\left(\frac{\partial}{\partial x_1}F_{\boldsymbol{\beta}}(\boldsymbol{x})\right)\tanh'(0.5F_{\boldsymbol{\beta}}(\boldsymbol{x}))$ is an even function in $x_1$. When $\beta_2^*>0$ and $x_2>0$, the difference term of the density function:

$$f\left(x_1-t\beta_1^*,x_2-\beta_2^*,x_3,\ldots,x_d\right)-f\left(x_1-t\beta_1^*,x_2+\beta_2^*,x_3,\ldots,x_d\right)>0,$$
$$f\left(x_1+t\beta_1^*,x_2-\beta_2^*,x_3,\ldots,x_d\right)-f\left(x_1+t\beta_1^*,x_2+\beta_2^*,x_3,\ldots,x_d\right)>0,$$

and $x_2\tanh'(0.5F_{\boldsymbol{\beta}}(\boldsymbol{x}))>0$. Moreover, $\frac{\partial}{\partial x_1}F_{\boldsymbol{\beta}}(\boldsymbol{x})\geq0$ and it is strictly positive as a function of $x_1$ on a subset of $(0,\infty)$ with a positive measure from Corollary D.1 . We thus conclude that $\frac{\partial}{\partial t}\beta_2^+(t)>0$ and $\beta_2^+>0$. $\qquad\square$

**Lemma D.3** (Fixed Point Structure in span($\boldsymbol{\beta}^*$))**.** *Suppose that the density function $f$ satisfies the regularity condition, $\boldsymbol{0},\boldsymbol{\beta}^*$ and $-\boldsymbol{\beta}^*$ are the only fixed points of the least-squares EM update in span($\boldsymbol{\beta}^*$).*

*Proof.* Assume $\sigma=1$. By Lemma 4.4, span($\boldsymbol{\beta}^*$) is an invariant subspace. We only need to consider $\beta_1^+$, which makes the problem one dimensional.

$$\beta_1^+=\int_{\boldsymbol{x}}\frac{1}{C_g}\exp\left(-g\left(\|(x_1-\|\boldsymbol{\beta}^*\|_2,x_2,\ldots,x_d)\|_2\right)\right)x_1\tanh(0.5F_{\boldsymbol{\beta}}(\boldsymbol{x}))\,\mathrm{d}\boldsymbol{x}.\qquad(25)$$

$\boldsymbol{0}$ is a trivial fixed point and we assume that $\|\boldsymbol{\beta}\|>0$ in the following. Conditioning on $x_2,\ldots,x_d$, $g\left(\frac{1}{\sigma}\|(x_1,x_2,\ldots,x_d)\|_2\right)$ is an even convex function in $x_1$, and it is strictly increasing when $x_1\geq0$. Theorem 4.1 tells us that:

$$\left|\int_{x_1}\frac{1}{C_{2:d}}\exp\left(-g\left(\|(x_1-\|\boldsymbol{\beta}^*\|_2,x_2,\ldots,x_d)\|_2\right)\right)x_1\tanh(0.5F_{\boldsymbol{\beta}}(\boldsymbol{x}))\,\mathrm{d}x_1-\|\boldsymbol{\beta}^*\|_2\right|$$

$$\leq\kappa_{2:d}(\|\boldsymbol{\beta}\|_2,\|\boldsymbol{\beta}^*\|_2,\sigma)\big|\|\boldsymbol{\beta}\|_2-\|\boldsymbol{\beta}^*\|_2\big|,$$

for some $\kappa_{2:d}(\|\boldsymbol{\beta}\|_2,\|\boldsymbol{\beta}^*\|_2,\sigma)\in(0,1)$. $C_{2:d}$ is the normalization factor for the density that is proportional to

$$\exp\left(-g\left(\|(x_1-\|\boldsymbol{\beta}^*\|_2,x_2,\ldots,x_d)\|_2\right)\right)$$

conditioning on $x_2,\ldots,x_d$. Now integrating over $x_2,\ldots,x_d$, we get $\left|\beta_1^+-\|\boldsymbol{\beta}^*\|_2\right|<\big|\|\boldsymbol{\beta}\|_2-\|\boldsymbol{\beta}^*\|_2\big|$ for all $\|\boldsymbol{\beta}\|_2>0$. The conclusion follows. $\qquad\square$

**Establishing Local Convergence** In the following, we denote $B(\boldsymbol{\beta}^*, \sigma)$ as the bound for the least-squares EM update. By Cauchy-Schwartz, we know that

$$\|\beta_i^+\|_2 \le \sqrt{\mathbb{E}_{\boldsymbol{X} \sim f_{\boldsymbol{\beta}^*,\sigma}} X_i^2} \quad \forall i.$$

Since the least-squares EM update is a two dimensional object, we can bound $\boldsymbol{\beta}^+$ by

$$\sqrt{\mathbb{E}_{\boldsymbol{X} \sim f_{\boldsymbol{\beta}^*,\sigma}} (X_1^2 + X_2^2)} := B(\boldsymbol{\beta}^*, \sigma).$$

**Lemma D.4** (Along $\widehat{\boldsymbol{\beta}}$). *Suppose that the density $f$ satisfies the regularity condition. We further assume that*

$$\sup_{t \in [0,1], \|\boldsymbol{\beta}\| \le B(\boldsymbol{\beta}^*, \sigma)} \int \frac{1}{\sigma^d C_g} \left( \exp\left( -g\left( \frac{1}{\sigma} \|(x_1 - \beta_1^*, x_2 - t\beta_2^*, \dots, x_d)\|_2 \right) \right) \right) \frac{\partial}{\partial x_2} [x_1 \tanh(0.5 F_{\boldsymbol{\beta},\sigma}(\boldsymbol{x}))] \, d\boldsymbol{x}$$

*is bounded by $D_1(\boldsymbol{\beta}^*, \sigma)$ in absolute value. When $\|\boldsymbol{\beta}\|_2 > 0$ and $\beta_1^* > 0$, the least-squares EM update satisfies: $\beta_1^+ > 0$ and*

$$|\beta_1^+ - \boldsymbol{\beta}^*| \le \kappa_1(\boldsymbol{\beta}^*, \boldsymbol{\beta}, \sigma) |\|\boldsymbol{\beta}\|_2 - \beta_1^*| + D_1(\boldsymbol{\beta}^*, \sigma)\beta_2^*.$$

*for some $\kappa_1(\boldsymbol{\beta}, \boldsymbol{\beta}^*, \sigma) \in (0, 1)$.*

*Proof.* Assume $\sigma = 1$. Recall that:

$$\beta_1^+ = \int_{x_1 \ge 0} \frac{1}{\sigma^d C_g} \left( \exp\left( -g\left( \|(x_1 - \beta_1^*, x_2 - \beta_2^*, x_3, \dots, x_d)\|_2 \right) \right) \right.$$

$$\left. + \exp\left( -g\left( \|(x_1 + \beta_1^*, x_2 - \beta_2^*, x_3, \dots, x_d)\|_2 \right) \right) \right) \cdot x_1 \tanh(0.5 F_{\boldsymbol{\beta}}(\boldsymbol{x})) \, d\boldsymbol{x}.$$

When $\|\boldsymbol{\beta}\|_2 > 0$, the integrand is strictly positive as $g\left( \frac{1}{\sigma} \|(x_1 + \|\boldsymbol{\beta}\|, x_2, \dots, x_d)\|_2 \right) - g\left( \frac{1}{\sigma} \|(x_1 - \|\boldsymbol{\beta}\|_2, x_2, \dots, x_d)\|_2 \right) > 0$ on the region where $x_1 > 0$. Thus

$$\beta_1^+ > 0.$$

This implies that $\|\boldsymbol{\beta}^+\|_2 > 0$ whenever $\|\boldsymbol{\beta}\|_2 > 0$.

Let us consider the following (slightly modified) quantity:

$$\beta_1^{++} = \int \frac{1}{C_g} \exp\left( -g\left( \|(x_1 - \beta_1^*, x_2, \dots, x_d)\|_2 \right) \right) x_1 \tanh(0.5 F_{\boldsymbol{\beta}}(\boldsymbol{x})) \, d\boldsymbol{x}$$

$$= \int_{x_{-1}} \int_{x_1} \frac{1}{C_g} \exp\left( -g\left( \|(x_1 - \beta_1^*, x_2, \dots, x_d)\|_2 \right) \right) x_1 \tanh(0.5 F_{\boldsymbol{\beta}}(\boldsymbol{x})) \, dx_1 \, dx_{-1}.$$

It is easy to see that conditioning on $x_2, \dots, x_d$, the inner integral is a one-dimensional least-squares EM operator with current estimate $\|\boldsymbol{\beta}\|_2$ and the true parameter $\beta_1^*$. Applying Theorem 4.1, we have

$$\left| \int_{x_1} \frac{1}{C_{2:d}} \exp\left( -g\left( \|(x_1 - \beta_1^*, x_2, \dots, x_d)\|_2 \right) \right) x_1 \tanh(0.5 F_{\boldsymbol{\beta}}(\boldsymbol{x})) \, dx_1 - \beta_1^* \right|$$

$$\le \kappa_{2:d}(\min(\beta_1^*, \|\boldsymbol{\beta}\|_2), \sigma) |\|\boldsymbol{\beta}\|_2 - \beta_1^*|,$$

where $\kappa_{2:d}(\min(\beta_1^*, \|\boldsymbol{\beta}\|_2), \sigma) < 1$ is a contraction factor depending on $x_2, \dots, x_d$. Integrating over $x_2, \dots, x_d$, we obtain

$$|\beta_1^{++} - \beta_1^*| \le \kappa_1(\min(\beta_1^*, \|\boldsymbol{\beta}\|_2), \sigma) |\|\boldsymbol{\beta}\|_2 - \beta_1^*|. \tag{26}$$

Next we bound $\beta_1^+ - \beta_1^{++}$. The regularity condition allows us to change the order of differentiation and integral.

$$\beta_1^+ - \beta_1^{++}$$

$$= \frac{\partial}{\partial t} \left[ \int \frac{1}{C_g} \left( \exp\left( -g\left( \frac{1}{\sigma} \|(x_1 - \beta_1^*, x_2 - t\beta_2^*, \dots, x_d)\|_2 \right) \right) \right) x_1 \tanh(0.5 F_{\boldsymbol{\beta}}(\boldsymbol{x})) \, d\boldsymbol{x} \right] |_{t \in (0,1)}$$

$$= \beta_2^* \left[ \int \frac{1}{C_g} \left( \exp\left( -g\left( \frac{1}{\sigma} \|(x_1 - \beta_1^*, x_2 - t\beta_2^*, \dots, x_d)\|_2 \right) \right) \right) \frac{\partial}{\partial x_2} [x_1 \tanh(0.5 F_{\boldsymbol{\beta}}(\boldsymbol{x}))] \, d\boldsymbol{x} \right]$$

$$\le D_1(\boldsymbol{\beta}^*, \sigma)\beta_2^*, \tag{27}$$

where in the last step, we used the assumption that the integral is uniformly bounded by $D_1(\boldsymbol{\beta}^*, \sigma)$, which only depends on $\boldsymbol{\beta}^*$ and $\sigma$ (This assumption is usually satisfied by the regularity condition.)

Combining (26) and (27), we can bound $\beta_1^+ - \beta_1^*$ as follows:

$$
\begin{aligned}
|\beta_1^+ - \beta_1^*| &= |\beta_1^{++} - \beta_1^* + \beta_1^+ - \beta_1^{++}| \\
&\leq |\beta_1^{++} - \beta_1^*| + |\beta_1^+ - \beta_1^{++}| \\
&\leq \kappa_1(\min(\beta_1^*, \|\boldsymbol{\beta}\|), \sigma)|\|\boldsymbol{\beta}\|_2 - \beta_1^*| + D_1(\boldsymbol{\beta}^*, \sigma)\beta_2^*,
\end{aligned}
\tag{28}
$$

and the conclusion follows. $\qquad\square$

**Lemma D.5** (Orthogonal to $\boldsymbol{\beta}^*$). *Assume that the density function $f$ satisfies the regularity condition. We further assume that*

$$
\sup_{t \in [0,1], \|\boldsymbol{\beta}\| \in B(\boldsymbol{\beta}^*, \sigma)} \int \frac{1}{\sigma^d C_g} \exp\left(g\left(\frac{1}{\sigma}\|(x_1 - \beta_1^*, x_2 - t\beta_2^*, \dots, x_d)\|_2\right)\right) \left|x_2 \frac{\partial}{\partial x_2}[0.5 F_{\boldsymbol{\beta}, \sigma}(\boldsymbol{x})]\right| \mathrm{d}\boldsymbol{x}
$$

*is uniformly bounded by $D_2(\boldsymbol{\beta}^*, \sigma)$. The following holds:*

$$
|\beta_2^* - \beta_2^+| \leq \kappa_2(\boldsymbol{\beta}, \boldsymbol{\beta}^*, \sigma)\beta_2^* + D_2(\boldsymbol{\beta}^*, \sigma)\beta_2^*
$$

*for some $\kappa_2(\boldsymbol{\beta}, \boldsymbol{\beta}^*, \sigma) \in (0, 1)$.*

*Proof.* Assume $\sigma = 1$. Recall that

$$
\beta_2^+ = \int \frac{1}{\sigma^d C_g} \exp\left(-g\left(\frac{1}{\sigma}\|(x_1 - \beta_1^*, x_2 - \beta_2^*, \dots, x_d)\|_1\right)\right) \cdot x_2 \tanh\left(0.5 F_{\boldsymbol{\beta}}(\boldsymbol{x})\right) \mathrm{d}\boldsymbol{x}.
$$

Consider the following quantity:

$$
\beta_2^{++} := \int \frac{1}{\sigma^d C_g} \exp\left(-g\left(\frac{1}{\sigma}\|(x_1 - \beta_1^*, x_2, \dots, x_d)\|_2\right)\right) \cdot x_2 \tanh\left(0.5 F_{\boldsymbol{\beta}}(\boldsymbol{x})\right) \mathrm{d}\boldsymbol{x} = 0,
$$

since the integrand is an odd function in $x_2$. Using the mean value theorem, we have:

$$
\beta_2^+ = \frac{\partial}{\partial t} \int \frac{1}{\sigma^d C_g} \exp\left(-g\left(\frac{1}{\sigma}\|(x_1 - \beta_1^*, x_2 - t\beta_2^*, \dots, x_d)\|_2\right)\right) x_2 \tanh\left(0.5 F_{\boldsymbol{\beta}}(\boldsymbol{x})\right) \mathrm{d}\boldsymbol{x} \Big|_{t \in [0,1]}.
$$

Under the regularity condition, we can interchange the order of differentiation and the integral:

$$
\begin{aligned}
\beta_2^+ = &\beta_2^* \cdot \int \frac{1}{C_g} \exp\left(-g\left(\frac{1}{\sigma}\|(x_1 - \beta_1^*, x_2 - t\beta_2^*, \dots, x_d)\|_2\right)\right) \tanh\left(0.5 F_{\boldsymbol{\beta}}(\boldsymbol{x})\right) \mathrm{d}\boldsymbol{x} \\
&+ \beta_2^* \cdot \int \frac{1}{C_g} \exp\left(-g\left(\frac{1}{\sigma}\|(x_1 - \beta_1^*, x_2 - t\beta_2^*, \dots, x_d)\|_2\right)\right) x_2 \frac{\partial}{\partial x_2}[0.5 F_{\boldsymbol{\beta}}(\boldsymbol{x})] \mathrm{d}\boldsymbol{x}.
\end{aligned}
\tag{29}
$$

Let us define:

$$
\kappa_2(\|\boldsymbol{\beta}\|, \boldsymbol{\beta}^*, \sigma) := \sup_{t \in (0,1)} \int \frac{1}{\sigma^d C_g} \exp\left(-g\left(\frac{1}{\sigma}\|(x_1 - \beta_1^*, x_2 - t\beta_2^*, \dots, x_d)\|_2\right)\right) \tanh\left(0.5 F_{\boldsymbol{\beta}, \sigma}(\boldsymbol{x})\right) \mathrm{d}\boldsymbol{x}.
$$

The first term for $\beta_2^+$ (29) is non-negative and is bounded by $\kappa_2(\boldsymbol{\beta}, \boldsymbol{\beta}^*, \sigma)\beta_2^*$ and the second term of $\beta_2^+$ (29) is bounded by $D_2(\boldsymbol{\beta}^*, \sigma)\beta_2^*$ by the assumption. The conclusion follows. $\qquad\square$

**Proposition D.6** (Local Quantitative bound). *Suppose that the density function $f$ satisfies the regularity condition, and further assume that the conditions in Lemma D.4 and D.5 holds. Then there exists $H(\boldsymbol{\beta}^*, \sigma)$ such that the following holds:*

$$
\|\boldsymbol{\beta}^+ - \boldsymbol{\beta}^*\|_2^2 \leq \kappa(\boldsymbol{\beta}, \boldsymbol{\beta}^*, \sigma)^2 \|\boldsymbol{\beta} - \boldsymbol{\beta}^*\|_2^2 + H(\boldsymbol{\beta}^*, \sigma)\sin(\angle(\boldsymbol{\beta}, \boldsymbol{\beta}^*)).
\tag{30}
$$

*where $\kappa(\boldsymbol{\beta}, \boldsymbol{\beta}^*, \sigma) < 1$.*

*Proof.* Since the least-squares EM update is bounded, we can assume that $\|\boldsymbol{\beta}\|_2$ is bounded without loss of generality. Furthermore, $\|\boldsymbol{\beta}\|_2 \le B(\boldsymbol{\beta}^*, \sigma)$.

$$
\begin{aligned}
\|\boldsymbol{\beta}^+ - \boldsymbol{\beta}^*\|_2^2 =& \|\beta_1^+ - \beta_1^*\|_2 + \|\beta_2^+ - \beta_2^*\|_2 \\
\le& (\kappa_1 |\|\boldsymbol{\beta}\|_2 - \beta_1^*| + D_1 \beta_2^*)^2 + (\kappa_2 \beta_2^* + D_2 \beta_2^*)^2 \\
=& \kappa_1^2 |\|\boldsymbol{\beta}\|_2 - \beta_1^*|^2 + 2 D_1 \beta_2^* \kappa_1 |\|\boldsymbol{\beta}\|_2 - \beta_1^*| + D_1^2 (\beta_2^*)^2 + \\
& \kappa_2^2 (\beta_2^*)^2 + 2 D_2 \beta_2^* \kappa_2 \beta_2^* + D_2^2 (\beta_2^*)^2 \\
\le& \kappa^2 \|\boldsymbol{\beta} - \boldsymbol{\beta}^*\|^2 + H(\boldsymbol{\beta}^*, \sigma) \sin(\angle(\boldsymbol{\beta}, \boldsymbol{\beta}^*)).
\end{aligned}
$$

In the last step, $\kappa = \max(\kappa_1, \kappa_2)$, and $H$ absorbs all the coefficient of $\beta_2^*$ in the cross term. It is easy to check that $H$ only depends on $\boldsymbol{\beta}^*$ and $\sigma$ as $\|\boldsymbol{\beta}\|_2$ is bounded by $B(\boldsymbol{\beta}^*, \sigma)$. □

**Establishing Local Stable Region**    Define

$$
\kappa_\tau := \max_{\boldsymbol{\beta} \in N_{\boldsymbol{\beta}^*}(\tau)} \kappa(\boldsymbol{\beta}, \boldsymbol{\beta}^*, \sigma) \tag{31}
$$

be the worst-case contraction factor in the $\tau$-neighborhood of $\boldsymbol{\beta}^*$, namely $N_{\boldsymbol{\beta}^*}(\tau) = \{\boldsymbol{\beta} : \|\boldsymbol{\beta} - \boldsymbol{\beta}^*\|_2 \le \tau \|\boldsymbol{\beta}^*\|_2\}$.

**Corollary D.7** (Local Stable Region). *Let $D$ be a positive number satisfying $D \le \frac{1}{2} \|\boldsymbol{\beta}^*\|_2$. Suppose that $\boldsymbol{\beta}^0 \in \mathbb{R}^d$ is such that (1) $\sin(\angle(\boldsymbol{\beta}^0, \boldsymbol{\beta}^*)) \le \frac{D^2(1-\kappa_{0.5}^2)}{H(\boldsymbol{\beta}^*, \sigma)}$ (2) $\|\boldsymbol{\beta}^0 - \boldsymbol{\beta}^*\|_2 \le D$, where $H(\boldsymbol{\beta}^*, \sigma)$ is defined in Corollary D.6. The following holds for the least-squares EM update $\boldsymbol{\beta}^1$ starting at $\boldsymbol{\beta}^0$:*

$$
\|\boldsymbol{\beta}^1 - \boldsymbol{\beta}^*\|_2 \le D, \quad \sin(\angle(\boldsymbol{\beta}^1, \boldsymbol{\beta}^*)) \le \frac{D^2(1-\kappa_{0.5}^2)}{H(\boldsymbol{\beta}^*, \sigma)}
$$

*Proof.* Applying Proposition D.6, we have

$$
\begin{aligned}
\|\boldsymbol{\beta}^1 - \boldsymbol{\beta}^*\|^2 \le& \kappa(\boldsymbol{\beta}^0, \boldsymbol{\beta}^*, \sigma)^2 \|\boldsymbol{\beta} - \boldsymbol{\beta}^*\|_2^2 + H(\boldsymbol{\beta}^*, \sigma) \sin(\angle(\boldsymbol{\beta}, \boldsymbol{\beta}^*)) \\
\le& \kappa_{0.5}^2 D^2 + H(\boldsymbol{\beta}^*, \sigma) \frac{D^2(1-\kappa_{0.5}^2)}{H(\boldsymbol{\beta}^*, \sigma)} = D^2.
\end{aligned}
$$

On the other hand, by the angle decreasing property, $\sin(\angle(\boldsymbol{\beta}^1, \boldsymbol{\beta}^*)) \le \sin(\angle(\boldsymbol{\beta}, \boldsymbol{\beta}^*)) \le \frac{D^2(1-\kappa_{0.5}^2)}{H(\boldsymbol{\beta}^*, \sigma)}$, thus the corollary is proved. □

**Establishing Norm Incerasing Region**

**Lemma D.8.** *There exists $\theta > 0$ such that $\|\boldsymbol{\beta}^+\|_2 > \|\boldsymbol{\beta}\|_2$ whenever $\boldsymbol{\beta} \in N_0(\frac{1}{8} \|\boldsymbol{\beta}^*\|_2)$ and $\angle(\boldsymbol{\beta}^*, \boldsymbol{\beta}) \le \theta$.*

*Proof.* Let us recall the expression for $\beta_1^+$ and $\beta_1^{++}$ (defined in Lemma D.4):

$$
\begin{aligned}
\beta_1^+ =& \int_{x_{-1}} \int_{x_1} \frac{1}{\sigma^d C_g} \exp\left(-g\left(\frac{1}{\sigma} \|(x_1 - \beta_1^*, x_2 - \beta_2^*, \dots, x_d)\|_2\right)\right) \\
& \cdot x_1 \tanh(0.5 F_{\boldsymbol{\beta}, \sigma}(\boldsymbol{x})) \, dx_1 \, dx_{-1}, \\
\beta_1^{++} =& \int_{x_{-1}} \int_{x_1} \frac{1}{\sigma^d C_g} \exp\left(-g\left(\frac{1}{\sigma} \|(x_1 - \beta_1^*, x_2, \dots, x_d)\|_2\right)\right) \\
& \cdot x_1 \tanh(0.5 F_{\boldsymbol{\beta}, \sigma}(\boldsymbol{x})) \, dx_1 \, dx_{-1}.
\end{aligned}
$$

$\beta_1^{++}$ is a modified iterate compared to $\beta_1^+$, and their discrepancy becomes smaller and smaller as $\angle(\boldsymbol{\beta}^*, \boldsymbol{\beta})$ goes to 0 ($\beta_1^{++} = \beta_1^+$ when $\angle(\boldsymbol{\beta}^*, \boldsymbol{\beta}) = 0$). For the modified iterate $\beta_1^{++}$, it has a fixed point $\|\boldsymbol{\beta}^*\|_2 \cos(\angle(\boldsymbol{\beta}^*, \boldsymbol{\beta}))$ along the $\widehat{\boldsymbol{\beta}}$ direction. Moreover, $\beta_1^{++}$ has the following two properties inherited from the structure of an one dimensional update:

- $\beta_1^{++}$ is increasing in $\|\boldsymbol{\beta}\|_2$;

Figure 2: The shaded region $S(\theta)$ is where $\beta_1^{++} > \|\boldsymbol{\beta}\|_2$.

- $\beta_1^{++} > \|\boldsymbol{\beta}\|_2$ whenever $\|\boldsymbol{\beta}\|_2 < \|\boldsymbol{\beta}^*\|_2 \cos(\angle(\boldsymbol{\beta}^*, \boldsymbol{\beta}))$.

In Figure 2, we illustrate the norm increasing region for $\beta_1^{++}$:
$$S(\theta) := \left\{ \boldsymbol{\beta} : \beta_1^{++} > \|\boldsymbol{\beta}\|_2, \angle(\boldsymbol{\beta}^*, \boldsymbol{\beta}) \leq \theta \right\}.$$

By the continuity of the least-squares EM update, it is easy to see that the norm increasing region for $\beta_1^+$:
$$\widetilde{S}(\theta) := \left\{ \boldsymbol{\beta} : \beta_1^+ > \|\boldsymbol{\beta}\|_2, \angle(\boldsymbol{\beta}^*, \boldsymbol{\beta}) \leq \theta \right\}$$
is close to $S(\theta)$ when $\theta$ is small (note that $S(\theta) = \widetilde{S}(\theta)$ when $\theta = 0$). Since for some $\Theta_0 > 0$, $N_0(1/8\|\boldsymbol{\beta}^*\|_2) \cap \{\boldsymbol{\beta} : \angle(\boldsymbol{\beta}^*, \boldsymbol{\beta}) \leq \theta\} \subseteq S(\Theta_0)$, i.e, $(\Theta_0)$ contains a (bounded) cone-shape region. We conclude that for sufficiently small $\theta$, $N_0(1/8\|\boldsymbol{\beta}^*\|_2) \cap \{\boldsymbol{\beta} : \angle(\boldsymbol{\beta}^*, \boldsymbol{\beta}) \leq \theta\} \subseteq \widetilde{S}(\theta)$. In other words, $\widetilde{S}(\theta)$ contains all those $\boldsymbol{\beta}$, whose angle with $\boldsymbol{\beta}^*$ is less than $\theta$ and whose norm is less than $\frac{1}{8}\|\boldsymbol{\beta}^*\|_2$. $\square$

**Proposition D.9.** *Suppose $\boldsymbol{\beta}^t \subseteq N_0(\frac{1}{8}\|\boldsymbol{\beta}^*\|_2)$ is a sequence of least-squares EM iterates and $\boldsymbol{\beta}^t \neq 0$ for all $t$, then it is impossible that $\lim_t \boldsymbol{\beta}^t = \mathbf{0}$.*

*Proof.* We argue by contradiction. By the angle decreasing property of the iterates, there exists $\theta > 0$ and $T$, such that $\angle(\boldsymbol{\beta}^*, \boldsymbol{\beta}^t) \leq \theta$ for all $t \geq T$. By Lemma D.8, we know that $\|\boldsymbol{\beta}^{t+1}\|_2 > \|\boldsymbol{\beta}^t\|_2 \geq \|\boldsymbol{\beta}^T\|_2 > 0$ for all $t \geq T$. Thus, the norm of the iterates is lower bounded by a positive number and it is impossible for the iterates to converge to $\mathbf{0}$. $\square$

## D.3 Proof of Lemma 4.3

For readability we restate the lemma below.

**Lemma 4.3** (Non-contraction in $\ell_2$). *Consider a log-concave density of the form $g(\boldsymbol{x}) \propto \|\boldsymbol{x}\|_2^r$ with $r \geq 1$. When $r \in [1,2]$, $\mathbf{0}$ is the only fixed point of LS-EM in the direction ortoghonal to $\boldsymbol{\beta}^*$. When $r \in (2,\infty)$, there exists a fixed point other than $\mathbf{0}$ in the orthogonal direction. Consequently, when $r > 2$, there exists $\boldsymbol{\beta}$ such that $\|M(\boldsymbol{\beta}^*, \boldsymbol{\beta}) - \boldsymbol{\beta}^*\|_2 > \|\boldsymbol{\beta} - \boldsymbol{\beta}^*\|_2$.*

*Proof.* Assume $\sigma = 1$. $\beta_1^+$ is an increasing function in $\|\boldsymbol{\beta}\|_2$ by Lemma D.10. Let us understand the derivative of $\frac{\partial \beta_1^+}{\partial \|\boldsymbol{\beta}\|_2}\big|_{\|\boldsymbol{\beta}\|_2=0}$ in (36) when $\angle(\boldsymbol{\beta}, \boldsymbol{\beta}^*) = \frac{\pi}{2}$. The expression is the following:

$$D_1(\|\boldsymbol{\beta}^*\|_2, f) := \frac{\partial \beta_1^+}{\partial \|\boldsymbol{\beta}\|_2}\big|_{\|\boldsymbol{\beta}\|_2=0} \tag{32}$$

$$= \int_{\boldsymbol{x}} \frac{1}{C_g} \exp\left(-g\left(\|(x_1, x_2 - \|\boldsymbol{\beta}^*\|, \ldots, x_d)\|_2\right)\right) x_1 \frac{\partial}{\partial x_1} g\left(\|(x_1, \ldots, x_d)\|_2\right) \mathrm{d}\boldsymbol{x}$$

$$= \int_{\boldsymbol{x}} \frac{1}{C_g} \exp(-g(\|(x_1, x_2 - \|\boldsymbol{\beta}^*\|, \ldots, x_d)\|_2)) x_1 \frac{\partial}{\partial x_1} g(\|(x_1, \ldots, x_d)\|_2) \mathrm{d}\boldsymbol{x}. \tag{33}$$

By a simple calculation, we have that when $\|\boldsymbol{\beta}^*\| = 0$, $D_1 = 1$. Let us further take the derivative with respect to $\|\boldsymbol{\beta}^*\|_2$:

$$\frac{d}{d\|\boldsymbol{\beta}^*\|_2} D_1(\|\boldsymbol{\beta}^*\|_2, f) = \mathbb{E}_{\boldsymbol{X} \sim f_{\boldsymbol{\beta}^*}} X_1 \frac{\partial^2}{\partial X_1 \partial X_2} g(\|(X_1, \dots, X_d)\|_2). \tag{34}$$

Here $f_{\boldsymbol{\beta}^*}(\boldsymbol{x}) = \frac{1}{C_g} \exp(-g(\|(x_1, x_2 - \beta_2^*, x_3, \dots, x_d)\|_2))$ (slightly different from the previous sections). In the special case where $g(\boldsymbol{x}) = c\|\boldsymbol{x}\|_2^r$ for some $r \geq 1$ and $c > 0$,

$$\frac{d}{d\|\boldsymbol{\beta}^*\|_2} D_1(\|\boldsymbol{\beta}^*\|_2, f) = \mathbb{E}_{\boldsymbol{X} \sim f_{\boldsymbol{\beta}^*}} r(r-2) X_1^2 X_2 \|\boldsymbol{X}\|_2^{\frac{r}{2} - 2}. \tag{35}$$

Since the integrand is an odd function in $X_2$, Lemma C.4 tells us that the above derivative is positive when $r \geq 2$; and the above derivative is negative when $r < 2$. This implies that when $r < 2$, $D_1(\|\boldsymbol{\beta}^*\|_2, f) < 1$, and when $r > 2$, $D_1(\|\boldsymbol{\beta}^*\|_2, f) > 1$. In particular, when $r > 2$, there is a positive fixed point $\widetilde{\boldsymbol{\beta}}$ (i.e., $\widetilde{\boldsymbol{\beta}}_1^+ = \|\widetilde{\boldsymbol{\beta}}\|$) for the least-squares EM operator in the orthogonal axis. Using the dynamics of the one dimensional LS-EM operator, we have that whenever $\|\boldsymbol{\beta}\|_2 < \|\widetilde{\boldsymbol{\beta}}\|_2$, $\|\boldsymbol{\beta}^+ - \widetilde{\boldsymbol{\beta}}\|_2 < \|\boldsymbol{\beta} - \widetilde{\boldsymbol{\beta}}\|_2$ for some $\boldsymbol{\beta}$ in the orthogonal direction. Consequently, $\|\boldsymbol{\beta}^+ - \boldsymbol{\beta}^*\|_2 > \|\boldsymbol{\beta} - \boldsymbol{\beta}^*\|_2$. We have completed the proof of Lemma 4.3 $\qquad\square$

**Lemma D.10.** *Under the regularity condition, $\beta_1^+$ is a strictly increasing function of $\|\boldsymbol{\beta}\|_2$.*

*Proof.* Assume $\sigma = 1$. Note that $\beta_1^+$ is a function of $\|\boldsymbol{\beta}\|_2$ and $\|\boldsymbol{\beta}^*\|_2$. We are interested in how $\beta_1^+$ will change with respect to $\|\boldsymbol{\beta}\|_2$. Under the regularity condition, we can take the derivative with respect to $\|\boldsymbol{\beta}\|_2$, which gives

$$\frac{\partial \beta_1^+}{\partial \|\boldsymbol{\beta}\|_2}$$
$$= \int_{x_{-1}} \int_{x_1} \frac{1}{C_g} f(x_1 - \beta_1^*, x_2 - \beta_2^*, x_3, \dots, x_d) \cdot x_1 \frac{\partial}{\partial \|\boldsymbol{\beta}\|} F_{\boldsymbol{\beta}}(\boldsymbol{x}) \tanh'(0.5 F_{\boldsymbol{\beta}}(\boldsymbol{x})) \, \mathrm{d}x_1 \, \mathrm{d}x_{-1}$$
$$= \int_{x_{-1}} \int_{x_1 \geq 0} \left( \frac{1}{C_g} f(x_1 - \beta_1^*, x_2 - \beta_2^*, x_3, \dots, x_d) - \frac{1}{C_g} f(x_1 + \beta_1^*, x_2 - \beta_2^*, x_3, \dots, x_d) \right)$$
$$\cdot x_1 \frac{\partial}{\partial \|\boldsymbol{\beta}\|} F_{\boldsymbol{\beta}}(\boldsymbol{x}) \tanh'(0.5 F_{\boldsymbol{\beta}}(\boldsymbol{x})) \, \mathrm{d}x \, \mathrm{d}x_{-1}.$$

We note that when $x_1 > 0$ and $\beta_1^* > 0$, $x_1 \tanh'(0.5 F_{\boldsymbol{\beta}}(\boldsymbol{x})) > 0$ and $f(x_1 - \beta_1^*, x_2 - \beta_2^*, x_3, \dots, x_d) > f(x_1 + \beta_1^*, x_2 - \beta_2^*, x_3, \dots, x_d)$. Meanwhile, $\frac{\partial}{\partial \|\boldsymbol{\beta}\|} F_{\boldsymbol{\beta}}(\boldsymbol{x})$ is positive on a subset of $(0, \infty)$ with positive measure by Corollary D.1. Therefore, $\frac{\partial \beta_1^+}{\partial \|\boldsymbol{\beta}\|_2} > 0$, and $\beta_1^+$ is an increasing function in $\|\boldsymbol{\beta}\|$. $\qquad\square$

## E  Regularity Condition

The regularity condition is a technical condition that makes changing the order of differentiation and integration valid. Formally, let us first recall the measure theory statement of Leibniz's integral rule

**Proposition E.1** (Theorem 16.8 of [7]). *Let $S$ be an open subset of $\mathbb{R}$, and $\Omega$ be a measure space. Suppose $f : S \times \Omega \to \mathbb{R}$ satisfies the following conditions: (1) $f(s, \omega)$ is a Lebesgue-integrable function of $\omega$ for each $s \in S$; (2) For almost all $\omega \in \Omega$, the derivative $f_s$ exists for all $s \in S$; (3) There is an integrable function $\theta : \Omega \to \mathbb{R}$ such that $|f_s(s, \omega)| \leq \theta(\omega)$ for all $s \in S$ and almost every $\omega \in \Omega$. It follows that:*

$$\frac{\mathrm{d}}{\mathrm{d}s} \int_\Omega f(s, \omega) \, \mathrm{d}\omega = \int_\Omega f_s(s, \omega) \, \mathrm{d}\omega. \tag{36}$$

In the above proposition, $S$ is the parameter space, and $\omega$ is the random variable. Recall the least-squares EM update function:

$$M(\boldsymbol{\beta}^*, \boldsymbol{\beta}) = \mathbb{E}_{\boldsymbol{X} \sim f_{\boldsymbol{\beta}^*, \sigma}} \boldsymbol{X} \tanh(0.5 F_{\boldsymbol{\beta}, \sigma}(\boldsymbol{X})),$$

$$F_{\boldsymbol{\beta}, \sigma}(\boldsymbol{X}) = g\left(\frac{1}{\sigma}\|X + \boldsymbol{\beta}\|_2\right) - g\left(\frac{1}{\sigma}\|X - \boldsymbol{\beta}\|_2\right).$$

Using the rotation invariant property of the distribution, we adopt a local orthogonal basis as in Lemma 4.4. The above two functions are equivalent to the following:

$$M(\boldsymbol{\beta}^*, \boldsymbol{\beta}) = \int_{\boldsymbol{x}} \frac{1}{\sigma^d C_g} \exp\left(-g\left(\|(x_1 - \beta_1^*, x_2 - \beta_2^*, x_3, \ldots, x_d)\|_2\right)\right) \boldsymbol{x} \tanh(0.5 F_{\boldsymbol{\beta}, \sigma}(\boldsymbol{x})) \, d\boldsymbol{x},$$

$$F_{\boldsymbol{\beta}, \sigma}(\boldsymbol{x}) = g\left(\frac{1}{\sigma}\|(x_1 + \|\boldsymbol{\beta}\|_2, x_2, \ldots, x_d)\|_2\right) - g\left(\frac{1}{\sigma}\|(x_1 - \|\boldsymbol{\beta}\|_2, x_2, \ldots, x_d)\|_2\right).$$

Indeed, from the above representation for $M(\boldsymbol{\beta}^*, \boldsymbol{\beta})$, we can think of the $M$ function as a function in three variables: $\beta_1^*, \beta_2^*$ and $\|\boldsymbol{\beta}\|_2$. Define, for each $i = 1, \ldots, d$,

$$M_i(z_1, z_2, z_2)$$
$$:= \int_{\boldsymbol{x}} \frac{1}{\sigma^d C_g} \exp\left(-g\left(\frac{1}{\sigma}\|(x_1 - z_1, x_2 - z_2, x_3, \ldots, x_d)\|_2\right)\right) \cdot$$
$$x_i \tanh\left(0.5\left(g\left(\frac{1}{\sigma}\|(x_1 + z_3, x_2, \ldots, x_d)\|_2\right) - g\left(\frac{1}{\sigma}\|(x_1 - z_3, x_2, \ldots, x_d)\|_2\right)\right)\right) \, d\boldsymbol{x}.$$
(37)

The regularity condition for $f \in \mathcal{F}$ ensures that for each $i = 1, \ldots, d$, $j = 1, 2, 3$, the following holds:

$$\frac{\partial M(z_1, z_2, z_3)}{\partial z_j}$$
$$= \int_{\boldsymbol{x}} \frac{1}{\sigma^d C_g} \frac{\partial}{\partial z_j} \Big[ \exp\left(-g\left(\|(x_1 - z_1, x_2 - z_2, x_3, \ldots, x_d)\|_2\right)\right) \cdot$$
$$x_i \tanh\left(0.5\left(g\left(\frac{1}{\sigma}\|(x_1 + z_3, x_2, \ldots, x_d)\|_2\right) - g\left(\frac{1}{\sigma}\|(x_1 - z_3, x_2, \ldots, x_d)\|_2\right)\right)\right) \Big] \, d\boldsymbol{x}.$$

In other words, we can differentiate the least-squares EM update with the parameter by putting the differentiation operator inside the integral. Note that the main technique for analyzing the least-squares EM update is the sensitivity analysis, in which we regularly differentiate $M(z_1, z_2, z_3)$ with one of the parameters.

In view of Leibniz's rule:

- The integrand is bounded by $|x_i|$ since the $\tanh(\cdot)$ function is uniformly bounded by 1. $\mathbb{E}_{\boldsymbol{X} \sim f_{\boldsymbol{\beta}^*, \sigma}} |x_i| < \infty$ for all $i$.

- $g$ is a convex function on $\mathbb{R}^+$, therefore, it is differentiable on $\mathbb{R}^+$ except on a measure 0 set. $\|\cdot\|_2$ is differentiable except at the origin. Thus, by the composition rule, we infer that the integrand is differentiable with $z_j$ ($j = 1, 2, 3$) for almost all $\boldsymbol{x} \in \mathbb{R}^d$.

- When we differentiate $M(z_1, z_2, z_3)$ with $z_j$, the parameter space is bounded. We differentiate $M_1$ with respect to $z_1$ in Lemma D.4, and its value is taken between $\min(\|\boldsymbol{\beta}\|_2, \beta_1^*)$ and $\max(\|\boldsymbol{\beta}\|_2, \beta_1^*)$; We differentiate $M_2$ with respect to $z_1$ in Lemma D.2, and its value is taken between 0 and $\beta_1^*$; We differentiate $M_1$ with respect to $z_2$ in D.4, and its value is taken between 0 and $\beta_2^*$; We differentiate $M_2$ with respect to $z_2$ in Lemma D.5 and its value is between 0 and $\beta_2^*$; We differentiate $M_1$ with respect to $z_3$ and its value is between 0 and $\|\boldsymbol{\beta}\|_2$. Since $\|\boldsymbol{\beta}\|_2$ is bounded by a function of $\boldsymbol{\beta}^*$ and $\sigma$, the above parameter space $S_j(\boldsymbol{\beta}^*, \sigma)$ for $z_j$ ($j = 1, 2, 3$) is all bounded. Therefore, it suffices to ensure the integrability of the derivative (with respect to $z_j$).

Therefore, in order to verify the regularity condition for a log concave distribution, one needs to ensure the derivative of integrand in $M_i$ ($i = 1, 2$) with respect to $j$ ($j = 1, 2, 3$) is integrable over

the parameter space $S_j(\boldsymbol{\beta}^*, \sigma)$. It suffices to show the following quantity is finite:

$$\int_{\boldsymbol{x}} \frac{1}{\sigma^d C_g} \sup_{z_j \in S_j(\boldsymbol{\beta}^*, \sigma)} \left| \frac{\partial}{\partial z_j} \left[ \exp\left(-g\left(\|(x_1 - z_1, x_2 - z_2, x_3, \ldots, x_d)\|_2\right)\right) \cdot \right. \right.$$

$$\left. \left. x_i \tanh\left(0.5\left(g\left(\frac{1}{\sigma}\|(x_1 + z_3, x_2, \ldots, x_d)\|_2\right) - g\left(\frac{1}{\sigma}\|(x_1 - z_3, x_2, \ldots, x_d)\|_2\right)\right)\right) \right] \right| \mathrm{d}\boldsymbol{x}$$

$$(38)$$

As an example, consider a general polynomial family, it is easy to verify that the above condition holds as a log concave distribution has finite moments of all the order.

## F  Finite Sample Analysis

In this section, we provide proofs for analysis of the LS-EM algorithm in the finite sample case. Proof of Proposition 5.1 is presented in Section F.1, which establishes an one-iteration bound. Proposition 5.2 and Proposition 5.3 are proved in Section F.2 for the global convergence. In Section F.3, we discuss the implication for some special distributions including Gaussian, Laplace and Logistic.

### F.1  Proof of Proposition 5.1

**Proposition 5.1** (1-d Finite Sample)**.** *Suppose the density function $f \in \mathcal{F}$ satisfies the regularity condition. With $\beta \in \mathbb{R}$ being the current estimate, the finite-sample LS-EM update* (8) *satisfies the following bound with probability at least $1 - \delta$:*

$$|\widetilde{\beta}^+ - \beta^*| \leq \kappa(\beta^*, \beta, \sigma) \cdot |\beta - \beta^*| + (\beta^* + C_f \sigma) \cdot O\left(\sqrt{\frac{1}{n} \log \frac{1}{\delta}}\right), \qquad (9)$$

*where $\kappa(\beta^*, \beta, \sigma)$ is contraction factor defined in Theorem 4.1 and $C_f$ is the Orlicz $\Psi_1$ norm (i.e., the sub-exponential parameter) of a random variable with density $f \in \mathcal{F}$.*

*Proof.* In the 1-d finite sample case, the least-squares EM update is

$$\widetilde{M}(\beta^*, \beta) = \frac{1}{n} \sum_{i=1}^{n} x^i \tanh(0.5 F_{\beta, \sigma}(x^i))$$

Since $|\tanh(\cdot)| \leq 1$, each summand $z^i := x^i \tanh(0.5 F_{\beta, \sigma}(x^i))$ is a sub-exponential random variable with $\Psi_1$ Orlicz norm upper bounded by $\beta^* + \sigma C_f$. This is because each $x_i \sim f_{\beta^*, \sigma}$ is a sub-exponential random variable (see Lemma F.2) with $\Psi_1$ Orlicz norm $O(\beta^* + \sigma C_f)$, where $C_f$ is the $\Psi_1$ Orlicz norm of a random variable with density $f$. Using Bernstein's inequality from Theorem F.3, we have

$$\mathbb{P}\{|\widetilde{M}(\beta^*, \beta) - M(\beta^*, \beta)| \geq t\} \leq 2\left[-cn \min\left(-\frac{t^2}{(\beta^* + \sigma C_f)^2}, \frac{t}{\beta^* + \sigma C_f}\right)\right].$$

Coupling with the one-step analysis for the population least-squares EM update in Theorem 4.1, we can bound the finite sample least-squares EM update as follows:

$$|\widetilde{M}(\beta, \beta^*) - \beta^*| \leq \kappa(\beta^*, \beta, \sigma)|\beta - \beta^*| + O\left(\sqrt{\frac{(\beta^* + \sigma C_f)^2}{n} \log \frac{1}{\delta}}\right)$$

with probability at least $1 - \delta$. $\qquad \square$

Let us recall the following equivalent definition for sub-exponential random variables:

**Lemma F.1** (Proposition 2.7.1 of [25])**.** *Let $X$ be a random variable in $\mathbb{R}$. $X$ is sub-exponential iff $\mathbb{E} \exp(|X|/K_3) \leq 2$ for some $K_3 > 0$.*

We first show a random variable with a symmetric log-concave density is necessarily sub-exponential.

**Lemma F.2.** *If a random variable $X$ has a log-concave density $f$ that is also an even function, then $X$ is sub-exponential, with the $\Psi_1$ Orlicz norm (sub-exponential norm) depending on $\log f$.*

*Proof.* For a general symmetric log-concave distribution with density $f(x) = \frac{1}{C_g} \exp(-g(|x|))$, the set of sub-differential of $g$: $\{\partial g(x) : x \in \mathbb{R}\}$ is non decreasing with respect to $x$. Moreover, the sub-differentials are non-negative when $x \geq 0$. Suppose that $\{\partial g(x) : x \in \mathbb{R}\}$ has an upper bound $C$, we can pick $M$ with $0 < M < C$ and $M \in \partial g(x_0)$ for some $x_0 > 0$. Otherwise, the $\{\partial g(x) : x \in \mathbb{R}\}$ does not have an upper bound and we can pick an arbitrary $M > 0$ such that $M \in \partial g(x_0)$ for some $x_0 > 0$. By definition of the sub-differential, we have

$$g(x) \geq g(x_0) + M(x - x_0).$$

Now let us compute the moment generating function: $\mathbb{E}_{X \sim f} \exp(\frac{1}{K}|X|)$ for some $K > \frac{1}{M}$.

$$
\begin{aligned}
\mathbb{E}_{X \sim f} \exp\left(\frac{1}{K}|X|\right) &= 2 \int_{x \geq 0} \frac{1}{C_g} \exp\left(\frac{1}{K}x\right) \exp(-g(x)) \, \mathrm{d}x \\
&\leq 2 \int_{x \geq 0} \frac{1}{C_g} \exp\left(\frac{1}{K}x\right) \exp(-g(x_0) - M(x - x_0)) \, \mathrm{d}x \\
&= 2 \int_{x \geq 0} \frac{1}{C_g} \exp(-g(x_0) + M x_0) \exp(-(M - \frac{1}{K})x) \, \mathrm{d}x \\
&= 2 \frac{1}{C_g} \exp(-g(x_0) + M x_0) \frac{1}{M - \frac{1}{K}} < \infty.
\end{aligned}
$$

Using the dominated convergence theorem, we know that

$$\lim_{K \to \infty} \mathbb{E}_{X \sim f} \exp\left(\frac{|X|}{K}\right) = 1,$$

thus, there exists some $K_0$ such that $\mathbb{E}_{X \sim f} \exp(\frac{|X|}{K}) \leq 2$. In particular, $X$ is sub-exponential (by Lemma F.1) with a finite $\|\cdot\|_{\Psi_1}$ Orlicz norm. $\square$

Having established the sub-exponential property of the log-concave distribution, we use $C_f$ to denote the $\Psi_1$ Orlicz norm for a log concave distribution $f$. With translation and scaling, it is not hard to see the $\Psi_1$ Orlicz norm for $f_{\beta^*, \sigma}$ is of the order $O(\beta^*) + \sigma C_f)$.

**Theorem F.3** (Bernstein's inequality Theorem 2.8.1 of [25]). *Let $X_1, \ldots, X_N$ be independent sub-exponential random variables. Then for every $t \geq 0$, we have*

$$\mathbb{P}\{|\sum_{i=1}^{N} X_i| \geq t\} \leq 2 \exp\left[-c \min\left(\frac{t^2}{\sum_{i=1}^{N} \|X_i\|_{\psi_1}^2}, \frac{t}{\max_i \|X_i\|_{\psi_1}}\right)\right],$$

*where $c > 0$ is an absolute constant.*

## F.2 Proofs of Proposition 5.2 and Proposition 5.3

For readability, we restate here again:

**Proposition 5.2** (First Stage: Escape from 0 and $\infty$). *Suppose the initial point $\beta^0$ is in $(0, 0.5\beta^*) \cup (1.5\beta^*, \infty)$. After $T = O\left(\log \frac{0.25\beta^*}{|\beta^0 - \beta^*|} / \log \kappa(\beta^*, \min(\beta^0, 0.5\beta^*), \sigma)\right)$ iterations, with $N/T = \Omega\left(\frac{(1 + C_f/\eta)^2}{(1 - \kappa(\beta^*, \min(\beta^0, 0.5\beta^*), \sigma))^2} \log \frac{1}{\delta}\right)$ fresh samples per iteration, LS-EM outputs a solution $\widetilde{\beta}^T \in (0.5\beta^*, 1.5\beta^*)$ with probability at least $1 - \delta \cdot O\left(\log \frac{0.25\beta^*}{|\beta^0 - \beta^*|} / \log \kappa(\beta^*, \min(\beta^0, 0.5\beta^*), \sigma)\right)$.*

**Proposition 5.3** (Second Stage: Local Convergence). *The following holds for any $\epsilon > 0$. Suppose $\beta^0 \in (0.5\beta^*, 1.5\beta^*)$. After $T = O\left(\log \epsilon / \log \kappa(\beta^*, 0.5\beta^*, \sigma)\right)$ iterations, with $N/T = \Omega\left(\frac{(\beta^* + C_f/\eta)^2}{\epsilon^2 (1 - \kappa(\beta^*, 0.5\beta^*, \sigma))^2} \log \frac{1}{\delta}\right)$ fresh samples per iteration, LS-EM outputs a solution $\widetilde{\beta}^T$ satisfying $|\widetilde{\beta}^T - \beta^*| \leq \epsilon \beta^*$ with probability at least $1 - \delta \cdot O\left(\log \epsilon / \log \kappa(\beta^*, 0.5\beta^*, \sigma)\right)$.*

*Proof.* The premise in Proposition 5.2 ensures that conditions in Lemma F.5 and Corollary F.6 hold, which guarantee that all the future iterates remain in $(\beta^0, \infty)$. There are two stages of analysis for the LS-EM algorithm in the finite sample case:

1. The initial $\beta^0$ is $\in (0, 0.5\beta^*)$ or the initial $\beta^0$ is $\in (1.5\beta^*, \infty)$. In this case, the iterates will get into the local stable region $(0.5\beta^*, 1.5\beta^*)$ quickly.

2. The iterates enters the stable region $(0.5\beta^*, 1.5\beta^*)$, and converge to an $\epsilon$-close estimate.

Let $\widetilde{\beta}^t$ denote the $t$-th iterate. The per iteration bound established in Proposition 5.1 says that with probability at least $1 - \delta$:

$$|\widetilde{\beta}^t - \beta^*| \leq \kappa(\beta^*, \widetilde{\beta}^{t-1}, \sigma)|\widetilde{\beta}^{t-1} - \beta^*| + O\left(\sqrt{\frac{(\beta^* + C_f\sigma)^2}{n} \log\frac{1}{\delta}}\right). \tag{39}$$

Let us first analyze the first stage:

In the case where $\beta^0 \in (0, 0.5\beta^*)$, the iterates contracts to $\beta^*$ initially by Lemma F.5. We use induction from step (39) to obtain:

$$|\widetilde{\beta}^t - \beta^*| \leq \kappa(\beta^*, \beta^0, \sigma)|\widetilde{\beta}^{t-1} - \beta^*| + \widetilde{O}\left(\sqrt{\frac{(\beta^* + C_f\sigma)^2}{n}}\right)$$

$$\leq \kappa(\beta^*, \beta^0, \sigma)^t|\beta^0 - \beta^*| + \frac{1}{1 - \kappa(\beta^*, \beta^0, \sigma)}O\left(\sqrt{\frac{(\beta^* + C_f\sigma)^2}{n} \log\frac{1}{\delta}}\right).$$

Under the assumption that the size of fresh samples per iteration satisfies $n = O\left(\frac{(1+C_f\eta)^2}{(1-\kappa(\beta^*,\beta^0,\sigma))^2}\right)$, we can guarantee that the accumulative statistical error is upper bounded: $\frac{1}{1-\kappa(\beta^*,\beta^0,\sigma)}O\left(\sqrt{\frac{C_f(\beta^*,\sigma)^2}{n} \log\frac{1}{\delta}}\right) \leq 0.25\beta^*$. Therefore, after $T = O\left(\frac{\log\frac{0.25\beta^*}{|\beta^0-\beta^*|}}{\log \kappa(\beta^*,\beta^0,\sigma)}\right)$ iterations, $|\widetilde{\beta}^t - \beta^*| < 0.5\beta^*$. The probability is at least $1 - \delta \cdot O\left(\frac{\log\frac{0.25\beta^*}{|\beta^0-\beta^*|}}{\log \kappa(\beta^*,\beta^0,\sigma)}\right)$ by a union bound.

In the case where $\beta^0 > 1.5\beta^*$, the sample complexity per iteration ensures that all future iterates are lower bounded by $0.5\beta^*$ (see the proof of corollary F.6.) We deduce the following:

$$|\widetilde{\beta}^t - \beta^*| \leq \kappa(\beta^*, 0.5\beta^*, \sigma)|\widetilde{\beta}^{t-1} - \beta^*| + \widetilde{O}\left(\sqrt{\frac{(\beta^* + C_f\sigma)^2}{n}}\right)$$

$$\leq \kappa(\beta^*, 0.5\beta^*, \sigma)^t|\beta^0 - \beta^*| + \frac{1}{1 - \kappa(\beta^*, 0.5\beta^*, \sigma)}O\left(\sqrt{\frac{(\beta^* + C_f\sigma)^2}{n} \log\frac{1}{\delta}}\right) \tag{40}$$

Again, the accumulative statistical error is bounded by $0.25\beta^*$. After $T = O\left(\frac{\log\frac{0.25\beta^*}{|\beta^0-\beta^*|}}{\log \kappa(\beta^*,0.5\beta^*,\sigma)}\right)$ iterations, $|\widetilde{\beta}^t - \beta^*| < 0.5\beta^*$. The probability is at least $1 - \delta \cdot O\left(\frac{\log\frac{0.25\beta^*}{|\beta^0-\beta^*|}}{\log \kappa(\beta^*,0.5\beta^*,\sigma)}\right)$ by a union bound.

Now let us analyze the second stage with the goal of achieving a relative error of $\epsilon$. Since the initial distance to $\beta^*$ is upper bounded by $0.5\beta^*$, it suffices to ensure the following:

$$0.5\kappa(\beta^*, 0, 5\beta^*, \sigma)^t\beta^* + \frac{1}{1 - \kappa(\beta^*, 0.5\beta^*, \sigma)}O\left(\sqrt{\frac{(\beta^* + C_f\sigma)^2}{n} \log\frac{1}{\delta}}\right) \leq \epsilon\beta^*,$$

so that the iterates get $\epsilon$-close to $\beta^*$. Again, the assumption on the sample complexity per iteration in Proposition 5.3 guarantees that the first part and the second part are both bounded by $0.5\epsilon\beta^*$. The proof is similar as before. $\square$

### F.2.1 Supporting Lemmas for Proposition 5.2 and Proposition 5.3

From the one-step analysis in the finite sample case as established in Proposition 5.1, we would like to determine the region of contraction to $\beta^*$ (i.e, $|\widetilde{\beta}^+ - \beta^*| < |\beta - \beta^*|$) with probability at least $1 - \delta$:

$$\mathcal{C}(f, \beta^*, \sigma) := \left\{ \beta : (1 - \kappa(\beta^*, \beta, \sigma))|\beta - \beta^*| > O\left( \sqrt{\frac{(\beta^* + C_f \sigma)^2}{n} \log \frac{1}{\delta}} \right) \right\}. \quad (41)$$

This region allows us to control the convergence rate for the iterates. Using the fact that $\kappa(\beta^*, \beta, \sigma)$ depends on $\min(\beta, \beta^*)$, a more explicit condition for the contraction region is the following:

$$(1 - \kappa(\beta^*, \beta, \sigma)) |\beta - \beta^*| \geq O\left( \sqrt{\frac{(\beta^* + C_f \sigma)^2}{n} \log \frac{1}{\delta}} \right) \quad \text{when } \beta < \beta^*; \quad (42)$$

$$(1 - \kappa(\beta^*, \beta^*, \sigma)) |\beta - \beta^*| \geq O\left( \sqrt{\frac{(\beta^* + C_f \sigma)^2}{n} \log \frac{1}{\delta}} \right) \quad \text{when } \beta > \beta^*. \quad (43)$$

Note that in (42), $\beta$ being close to $0$ or close to $\beta^*$ will make the left hand side vanish, thus we infer that the contraction region for $\beta \leq \beta^*$ is an open interval $(L_1, L_2)$. In (43), we infer that the contraction region for $\beta > \beta^*$ is an open interval $(R_1, \infty)$. We provide an illustration in Figure 3:

Figure 3: Contraction region: $(L_1, L_2)$ and $(R_1, \infty)$

**Lemma F.4** (Contraction implies Stability). *Suppose that $\widehat{\beta} \in \mathcal{C}(f, \beta^*, \sigma)$ and $\widehat{\beta} < \beta^*$. For all $\beta \in (\widehat{\beta}, 2\beta^* - \widehat{\beta})$, we have $\widetilde{\beta}^+ \in (\widehat{\beta}, 2\beta^* - \widehat{\beta})$.*

*Proof.* Using Proposition 5.1, we have

$$|\widetilde{\beta}^+ - \beta^*| \leq \kappa(\beta^*, \beta, \sigma)|\beta - \beta^*| + O\left( \sqrt{\frac{(\beta^* + C_f \sigma)^2}{n} \log \frac{1}{\delta}} \right)$$

$$\leq \kappa(\beta^*, \widehat{\beta}, \sigma)|\widehat{\beta} - \beta^*| + O\left( \sqrt{\frac{(\beta^* + C_f \sigma)^2}{n} \log \frac{1}{\delta}} \right) \quad (44)$$

$$\leq |\widehat{\beta} - \beta^*|, \quad (45)$$

where (44) follows from $\kappa(\beta^*, \beta, \sigma) \leq \kappa(\beta^*, \widehat{\beta}, \sigma)$ and $|\beta - \beta^*| \leq |\widehat{\beta} - \beta^*|$. Step (45) follows by the assumption that $\widehat{\beta} \in \mathcal{C}(f, \beta^*, \sigma)$ and (42). $\square$

**Lemma F.5.** *For every $\ell \in (0, 0.5\beta^*)$, suppose that $n = \widetilde{\Omega}\left( \frac{(1 + C_f / \eta)^2}{(1 - \kappa(\beta^*, \ell, \sigma))^2} \right)$, we have that both $\ell$ and $0.5\beta^*$ are in $\mathcal{C}(f, \beta^*, \sigma)$, where $\eta = \beta^* / \sigma$.*

*Proof.* For $\ell < 0.5\beta^*$ to be in the contraction region, a sufficient condition is the following:

$$0.5\beta^* (1 - \kappa(\beta^*, \ell, \sigma)) \geq O\left( \sqrt{\frac{(\beta^* + C_f \sigma)^2}{n} \log \frac{1}{\delta}} \right).$$

A little algebra shows that $n = \widetilde{\Omega}\left( \frac{(1 + C_f / \eta)^2}{(1 - \kappa(\beta^*, \ell, \sigma))^2} \right)$. Indeed, the above condition also implies that $0.5\beta^* \in \mathcal{C}(f, \beta^*, \sigma)$ since $\kappa(\beta^*, \ell, \sigma) > \kappa(\beta^*, 0.5\beta^*, \sigma)$. $\square$

**Corollary F.6.** *Let $\ell \in (0, 0.5\beta^*)$. Suppose that $n = \widetilde{\Omega}\left(\frac{(1+C_f/\eta)^2}{(1-\kappa(\beta^*, \ell, \sigma))^2}\right)$, we have that for all $\beta \in (\ell, \infty)$, $\widetilde{\beta}^+ \in (\ell, \infty)$.*

*Proof.* By Lemma F.5, $\ell \in \mathcal{C}(f, \beta^*, \sigma)$. By Lemma F.4, we have $\widetilde{\beta}^+ > \ell$ when $\beta \in (\ell, 2\beta^* - \ell)$. Therefore, it suffices to consider the case when $\beta > 2\beta^* - \ell$. From (1) the property of population least-squares EM update for $\beta^+$: $\beta^+ > \beta^*$ and (2) the intermediate result from Proposition 5.1 that with probability at least $1 - \delta$:

$$|\widetilde{\beta}^+ - \beta^+| \leq O\left(\sqrt{\frac{(\beta^* + C_f\sigma)^2}{n} \log \frac{1}{\delta}}\right),$$

it follows that

$$\begin{aligned}\widetilde{\beta}^+ \geq &\beta^+ - O\left(\sqrt{\frac{(\beta^* + C_f\sigma)^2}{n} \log \frac{1}{\delta}}\right) \\ > &\beta^* - O\left(\sqrt{\frac{(\beta^* + C_f\sigma)^2}{n} \log \frac{1}{\delta}}\right) > 0.5\beta^* > \ell.\end{aligned}$$

$\square$

## F.3 Finite-sample convergence guarantees for special cases

We have shown in Section C.2 that for Gaussian, Laplace and logistic distribution, the contraction factor takes the form $\kappa(\beta^*, \beta, \sigma) = \exp\left(-c\frac{\min(\beta, \beta^*)^{\gamma_f}}{\sigma^{\gamma_f}}\right)$, for some $\gamma_f \geq 1$ determined by the asymptotic growth of the log density.

In view of Propositions 5.2 and 5.3, we deduce the following overall convergence result:

**Corollary F.7** (Explicit Convergence Rate). *Suppose that log concave density $f$ satisfies the regularity condition, and the contraction ratio $\kappa(\beta^*, \beta, \sigma) = \exp\left(-\frac{\min(\beta, \beta^*)^{\gamma_f}}{\sigma^{\gamma_f}}\right)$. We run the LS-EM algorithm with a an initial point $\beta^0$. If $\beta^0$ falls in the local region of $\beta^*$:$(0.5\beta^*, 1.5\beta^*)$, the LS-EM algorithm outputs a solution $\widetilde{\beta}^T$ such that $|\widetilde{\beta}^T - \beta^*| \leq \epsilon\beta^*$ after $T = O\left(\log \epsilon/\eta^{\gamma_f}\right)$ iterations. The number of fresh samples required per iteration is: $N/T = \widetilde{\Omega}\left(\frac{(\beta^* + C_f/\eta)^2}{\epsilon^2\eta^{2\gamma_f}}\right)$. Otherwise, if the initial point $\beta^0 \in (0, 0.5\beta^*)$ or $(0.5\beta^*, \infty)$, the LS-EM algorithm will take an additional $T' = O\left(\frac{\log \frac{0.25\beta^*}{|\beta^0 - \beta^*|}}{\eta^{\gamma_f}}\right)$ iterations before the iterates enter the local region $(0.5\beta^*, 1.5\beta^*)$. The number of fresh samples required per iteration is $\widetilde{\Omega}\left(\frac{(1+C_f/\eta)^2}{\eta^{2\gamma_f}}\right)$.*

# G Model Mis-specification

In this section, we establish the robustness results for the LS-EM algorithm with a mis-specified distribution in 1-D. Proposition 6.1 is proved in Section G.1. In Section G.2, we present some numerical observations for the robustness of the LS-EM algorithm.

## G.1 Proof of Proposition 6.1

**Lemma G.1** (3 fixed points when misspecified). *Suppose that $f \in \mathcal{F}$ satisfy the regularity condition. We further assume in the region $\beta \geq 0$, the function $\widehat{F}_{\beta,\sigma}(x) := \widehat{g}\left(\frac{1}{\sigma}|x + \beta|\right) - \widehat{g}\left(\frac{1}{\sigma}|x - \beta|\right)$ is a concave function in $\beta$ for each $x \geq 0$, and*

$$\mathbb{E}_{X \sim f_{\beta^*}} X\widehat{g}'(X) > 1.$$

*The iterates of the LS-EM algorithm with mis-specified log-concave density $\widehat{g}$ converge to a non-zero $\overline{\beta}$ ( or $-\overline{\beta}$) from a non-zero random initialization.*

*Proof.* With mis-specified log density $\widehat{g}$, the next iterate is:

$$
\begin{aligned}
\widehat{\beta}^+ &= \mathbb{E}_{X \sim f_{\beta^*},\sigma} X \tanh(0.5 \widehat{F}_{\beta,\sigma}(X)) \\
&= \int_{x \geq 0} \left( \frac{1}{\sigma} f\left( \frac{x - \beta^*}{\sigma} \right) + \frac{1}{\sigma} f\left( \frac{x + \beta^*}{\sigma} \right) \right) x \tanh(0.5 \widehat{F}_{\beta,\sigma}(x)) \, \mathrm{d}x \\
&=: \widehat{M}(\beta^*, \beta).
\end{aligned}
$$

We state the key properties for the function $\widehat{M}(\beta^*, \cdot)$:

- $\widehat{M}(\beta^*, \cdot)$ is an increasing function in $\beta$: we utilize the convexity of $\widehat{g}$, similar to the proof of Lemma D.10;

- $\widehat{M}(\beta^*, \cdot)$ is a concave function in $\beta$: since $\widehat{F}_{\beta,\sigma}$ is a concave function in $\beta$ for every $x \geq 0$, so is $\tanh(0.5 \widehat{F}_{\beta,\sigma})$ which is a composition with a concave function $\tanh$ that is increasing on the non-negative part. $\widehat{M}(\beta^*, \cdot)$ is concave in $\beta$ since it is an intergal of concave functions of $\beta$;

- $\widehat{M}(\beta^*, 0) = 0$;

- $\frac{\partial}{\partial \beta} \widehat{M}(\beta^*, \beta) \mid_{\beta=0} > 1$: the assumption that $\mathbb{E}_{X \sim f_{\beta^*}} X \widehat{g}'(X) \geq 1$ is equivalent to $\frac{\partial \widehat{M}(\beta^*, \beta)}{\partial \beta} \mid_{\beta=0} > 1$;

- $\widehat{M}(\beta^*, \beta) - \beta \to -\infty$ as $\beta \to \infty$: since $\widehat{M}(\beta^*, \beta)$ is bounded.

The above three properties guarantees that on $(0, \infty)$, $\widehat{M}(\beta^*, \cdot)$ has a unique fixed point $\overline{\beta} > 0$ satisfying the following properties: (1) $\widehat{M}(\beta^*, \overline{\beta}) = \overline{\beta}$; (2) If $\beta \in (0, \overline{\beta})$, $\widehat{M}(\beta^*, \overline{\beta}) \in (\beta, \overline{\beta})$; (3) If $\beta \in (\overline{\beta}, \infty)$, $\widehat{M}(\beta^*, \overline{\beta}) \in (\overline{\beta}, \beta)$. Since $\widehat{M}(\beta^*, \cdot)$ is an odd function in $\beta$, we conclude that on $(-\infty, 0)$, $\widehat{M}(\beta^*, \cdot)$ has a unique fixed point $-\overline{\beta}$. In view of the above properties, we deduce that if an initial point is positive, it converges to $\overline{\beta}$; and if it is negative, it converges to $-\overline{\beta}$. $\qquad\square$

**Lemma G.2** (Error bound when misspecified). *Suppose that the assumption in Lemma G.1 holds and the function $x \tanh(0.5 \widehat{F}_{\beta,\sigma}(x))$ is L-Lipschitz. For any $\beta^0 \neq 0$, the LS-EM with misspecified log concave distribution $\widehat{f}$ will converge to a solution $\overline{\beta}$ ,and*

$$
|\overline{\beta} - \operatorname{sign}(\beta^0, \beta^*) \beta^*| \leq \frac{6\sigma}{1 - \kappa(\overline{\beta}, \beta^*, \sigma)},
$$

*where $\kappa(\overline{\beta}, \beta^*, \sigma) \in (0, 1)$ is defined in Theorem 4.1.*

*Proof.* The fixed point structure established in Lemma G.1 ensures that the iterates converge to either $\overline{\beta}$ or $-\overline{\beta}$, depending on the sign of $\beta^0$. $\overline{\beta}$ satisfies $\mathbb{E}_{X \sim f_{\beta^*},\sigma} X \tanh(0.5 \widehat{F}_{\overline{\beta},\sigma}(X)) = \overline{\beta}$. We can decompose the difference between $\overline{\beta}$ and $\beta^*$ in the following way (utilizing the consistency property of the LS-EM update):

$$
\begin{aligned}
\overline{\beta} - \beta^* &= \mathbb{E}_{X \sim f_{\beta^*},\sigma} X \tanh(0.5 \widehat{F}_{\overline{\beta},\sigma}(X)) - \mathbb{E}_{X \sim \widehat{f}_{\beta^*},\sigma} X \tanh(0.5 \widehat{F}_{\beta^*,\sigma}(X)) \\
&= \underbrace{\mathbb{E}_{X \sim f_{\beta^*},\sigma} X \tanh(0.5 \widehat{F}_{\overline{\beta},\sigma}(X)) - \mathbb{E}_{X \sim \widehat{f}_{\beta^*},\sigma} X \tanh(0.5 \widehat{F}_{\overline{\beta},\sigma}(X))}_{A} \\
&\quad + \underbrace{\mathbb{E}_{X \sim \widehat{f}_{\beta^*},\sigma} X \tanh(0.5 \widehat{F}_{\overline{\beta},\sigma}(X)) - \mathbb{E}_{X \sim \widehat{f}_{\beta^*},\sigma} X \tanh(0.5 \widehat{F}_{\beta^*,\sigma}(X))}_{B}.
\end{aligned}
$$

Let us control $A$ and $B$ separately. The term $B$ is exactly the difference between a least-squares EM update and the true location parameter with the log-concave distribution $\widehat{f}$. Therefore, Theorem 4.1 tells us that

$$
|B| \leq \kappa_{\widehat{f}}(\overline{\beta}, \beta^*, \sigma) |\beta - \beta^*|.
$$

for some $\kappa_{\widehat{f}}(\overline{\beta}, \beta^*, \sigma) \in (0, 1)$. For term $A$, we note that the integrand is $L$-Lipschitz (by the assumption), it can be bounded by the Wasserstein distance between two distributions as follows:

$$A \leq L \cdot D_W \left( \frac{1}{\sigma} f \left( \frac{1}{\sigma} (\cdot - \beta^*) \right), \frac{1}{\sigma} \widehat{f} \left( \frac{1}{\sigma} (\cdot - \beta^*) \right) \right).$$

Here we use $D_W$ to denote the Wasserstein distance. By scaling and translation, we have:

$$D_W \left( \frac{1}{\sigma} f \left( \frac{1}{\sigma} (\cdot - \beta^*) \right), \frac{1}{\sigma} \widehat{f} \left( \frac{1}{\sigma} (\cdot - \beta^*) \right) \right) = \sigma D_W(f, \widehat{f}),$$

where $f$ and $\widehat{f}$ are two log-concave distribution with unit variance. Using the triangle inequality, it can be further bounded by:

$$\begin{aligned} D_W(f, \widehat{f}) &\leq D_W(f, \mathcal{N}(0, 1)) + D_W(\widehat{f}, \mathcal{N}(0, 1)) \\ &\leq 2 \sup_{f \in \mathcal{F}} D_W(f, \mathcal{N}(0, 1)). \end{aligned}$$

Now we can apply the classical Stein's method to bound the Wasserstein distance; in particular, we apply Proposition G.3 and obtain that:

$$\begin{aligned} D_W(f, \mathcal{N}(0, 1)) &\leq |\mathbb{E}_{W \sim f} W h(W) - h'(W)| \\ &\leq 2\mathbb{E}_{W \sim f} |W| + \sqrt{\frac{\pi}{2}} \\ &\leq 2\sqrt{\mathbb{E}_{W \sim f} W^2} + \sqrt{\frac{\pi}{2}} \leq 3. \end{aligned}$$

The last line follows since we assume $W \sim f$ has unit variance. Combining the bound on $A$ and $B$ together, we have proved the following:

$$|\overline{\beta} - \beta^*| \leq \kappa_{\widehat{f}}(\overline{\beta}, \beta^*, \sigma)|\overline{\beta} - \beta^*| + 6L\sigma,$$

and rearranging the inequality yields:

$$|\overline{\beta} - \beta^*| \leq \frac{6L\sigma}{1 - \kappa_{\widehat{f}}(\overline{\beta}, \beta^*, \sigma)}. \tag{46}$$

This completes the proof. $\qquad \square$

**Fitting with Gaussian**   Now let us consider a special case where $\widehat{f}$ is the Gaussian distribution. The *misspecified* variant EM update is the following:

$$\widehat{M}(\beta^*, \beta) = \mathbb{E}_{X \sim f_{\beta^*, \sigma}} X \tanh \left( \frac{\beta X}{\sigma^2} \right) \tag{47}$$

The conditions in Lemma G.2 satisfies. Furthermore, we can prove a lower bound for $\overline{\beta}$ in (46) when the SNR is high. This allows us to obtain a better error bound for controlling the distance between $\overline{\beta}$ and $\beta^*$. This is the content of Proposition 6.1, restated below.

**Proposition 6.1** (Fit with 2GMM). *Under the above one dimensional setting with Gaussian $\widehat{f}$, the following holds for some absolute constant $C_0 > 0$: If $\eta \geq C_0$, then the LS-EM algorithm with a non-zero initialization point $\beta^0$ converges to a solution $\overline{\beta}$ satisfying* $\operatorname{sign}(\overline{\beta}) = \operatorname{sign}(\beta^0)$ *and*

$$\left| \overline{\beta} - \operatorname{sign}(\beta^0 \beta^*) \beta^* \right| \leq 10\sigma.$$

*Proof.* The gradient with respect to $\beta = 0$ for $\widehat{M}(\beta^*, \beta)$ is $1 + \frac{(\beta^*)^2}{\sigma^2} > 1$. Meanwhile, it is easy to see that $\widehat{M}(\beta^*, \beta)$ is a concave function of $\beta$ on the region where $\beta \geq 0$. Moreover, $x \tanh(\frac{\beta x}{\sigma^2})$ is 1.5-Lipschitz as a function of $x$ for all $\beta \geq 0$. Lemma G.1 is applicable and it tells us that the mis-specified variant EM updates converge to a point $\overline{\beta}$ from a random non-zero initialization.

In the following: let $M_g(\beta^*, \beta)$ denote the least-squares EM update with the ground truth log-concave distribution: $0.5\mathcal{N}(\beta^*, \sigma^2) + 0.5\mathcal{N}(-\beta^*, \sigma^2)$. In Lemma G.2, we have proved an intermediate result (bound for term $A$) that:

$$|\widehat{M}(\beta^*, \beta) - M_g(\beta^*, \beta)| \leq 9\sigma, \tag{48}$$

Meanwhile, the convergence result from [10] (i.e., Corollary C.6) says that at $\beta = \frac{\beta^*}{2}$,

$$\beta^* - M_g(\beta^*, 0.5\beta^*) \leq 0.5\exp\left(-\frac{(\beta^*)^2}{8\sigma^2}\right)\beta^*. \tag{49}$$

Combining (48) and (49), We deduce that:

$$\begin{aligned}
\widehat{M}(\beta^*, 0.5\beta^*) \geq & M_g(\beta^*, 0.5\beta^*) - 9\sigma \\
\geq & \beta^*\left(1 - 0.5\exp\left(-\frac{(\beta^*)^2}{8\sigma^2}\right)\right) - 9\sigma \\
= & \beta^*\left(1 - 0.5\exp(-0.125\eta^2) - \frac{1}{\eta}\right).
\end{aligned}$$

When $\eta > C_0$ for some absolute constant $C_0 > 0$, we can show that $\beta^*\left(1 - 0.5\exp(-0.125\eta^2) - \frac{1}{\eta}\right) > 0.5\beta^*$. In particular, this implies that $\overline{\beta} > 0.5\beta^*$ by Lemma G.1. Therefore, the error bound in (46) can be further bounded by

$$|\overline{\beta} - \beta^*| \leq \frac{9\sigma}{1 - \exp\left(-0.125\eta^2\right)}.$$

The right hand side bound is smaller than $10\sigma$ when $\eta$ is large. This completes the proof of Proposition 6.1. $\qquad\square$

**Proposition G.3** (Wasserstein Distance Bound by Stein's Method [20]). *We have*

$$D_W(f, \mathcal{N}(0,1)) \leq \sup_{h \in \mathcal{F}} |\mathbb{E}_{W \sim f}[Wh(W) - h'(W)]|,$$

*where* $\mathcal{F} = \{h : \|h\| \leq 2, \|h'\| \leq \sqrt{\frac{\pi}{2}}, \|h''\| \leq 2\}$.

### G.2 General Observations

In Section G.1, the robustness results rely on the assumptions in Lemma G.1. In particular, we need the concavity of $\widehat{F}_{\beta,\sigma}$. This is a very restrictive condition. Consider the family of the log-concave distribution whose log density is of the form $g(x) \propto |x|^r$, $r \geq 1$. The concavity condition holds only when $r \leq 2$. However, the 3-fixed point structure still holds as along as the five properties in the proof for Lemma G.1 hold. Indeed we observe that the function $\beta \to \widehat{M}(\beta^*, \beta)$ is a concave function in $\beta$ even when $r > 2$ (see Section H.5).

Recall that the least-squares EM iterate with the misspecified distribution $\widehat{f}$ is the following:

$$\widehat{M}(\beta^*, \beta) = \mathbb{E}_{X \sim f_{\beta^*,\sigma}} x\tanh(0.5\widehat{F}_{\beta,\sigma}(X)), \tag{50}$$

where

$$\begin{aligned}
\widehat{g} &= \log\widehat{f} \\
\widehat{F}_{\beta,\sigma}(X) &= \widehat{g}\left(\frac{1}{\sigma}|X - \beta|\right) - \widehat{g}\left(\frac{1}{\sigma}|X + \beta|\right).
\end{aligned}$$

We can decompose the iterate $\widehat{M}(\beta^*, \beta)$ as follows:

$$\begin{aligned}
\widehat{M}(\beta^*, \beta) = & \mathbb{E}_{X \sim f_{\beta^*,\sigma}} x\tanh(0.5\widehat{F}_{\beta,\sigma}(x)) - \mathbb{E}_{X \sim \widehat{f}_{\beta^*,\sigma}} x\tanh(0.5\widehat{F}_{\beta,\sigma}(X)) \\
& + \mathbb{E}_{X \sim \widehat{f}_{\beta^*,\sigma}} x\tanh(0.5\widehat{F}_{\beta,\sigma}(X)), \tag{51}
\end{aligned}$$

where the first difference term is a *drift* term specifying the error due to the distribution misspecification and the second term is the ideal LS-EM update with $\widehat{f}$, which contracts to $\beta^*$ at a linear rate. We empirically observe that if we fit a lighter tail log-concave distribution $\widehat{f}$ compared to $f$, the drift term is positive (see Section H.5) and thus the least-squares EM iterate converges to some $\overline{\beta} > \beta^*$. By the triangle inequality,

$$\widehat{M}(\beta^*, \overline{\beta}) \leq \mathbb{E}_{X \sim f_{\beta^*, \sigma}} |X \tanh(0.5 \widehat{F}_{\overline{\beta}, \sigma}(X))| \leq \beta^* + \mathbb{E}_{x \sim f_{\beta^*, \sigma}} |X| \leq \beta^* + \sigma.$$

Thus the relative error is bounded by $\frac{\sigma}{\beta^*} = \frac{1}{\eta}$. When the SNR is large, the error is small. On the other hand, if we fit a heavier tail distribuiton $\widehat{f}$ compared to $f$, the drift is negative and the corresponding fixed point can be 0. This suggests a practical recipe: when one does not know the ground truth log-concave density, fit with a density that has a lighter tail. For instance, we can fit a Gaussian density when the ground truth is Laplace or logistic.

## H    Discussion and Numerical Experiments

In this section, we first examine the two key assumptions on the distribution class, namely rotation invariance and log-concavity, respectively in Sections H.1 and H.2. In particular, we discuss the obstacles of relaxing these two assumptions and provide numerical evidences. In Section H.3, we discuss the least-squares M-step being an approximate M-step in the classical EM algorithm. In Section H.4, we numerically verify the non $\ell_2$ convergence behavior for a general log-concave distributions. In Section H.5, we study the convergence behavior as well as the quality of the solution for the LS-EM with a mis-specified distribution.

### H.1    Assumption of Rotation Invariance

For the distribution class considered in (1), we assume the density function $f$ is of the form $\frac{1}{C_g} \exp(-g(\|\cdot\|))$. This automatically encodes the rotation invariance property of the distribution. Examining the analysis in detail, we find that the rotation invariance allows for the following property: for any orthonormal matrix $Q$, $g(\|Q(\boldsymbol{x})\|)$ is symmetric in $x_j$ for all $j \geq 3$. Consequently, the LS-EM updates for $j$-th coordinate ($j \geq 3$):

$$\mathbb{E}_{\boldsymbol{X} \sim f_{\boldsymbol{\beta}^*} \cdot Q} x_j \tanh(0.5(g(\|Q((x_1 + \|\boldsymbol{\beta}\|, x_2, \ldots, x_d))\|) - g(\|Q((x_1 - \|\boldsymbol{\beta}\|, x_2, \ldots, x_d))\|)))$$

vanishes. In general, if we want to relax the rotation invariance property by assuming $g$ as a function of $\boldsymbol{x}$, the property can be generalized as follows: for any $\boldsymbol{u}, \boldsymbol{v} \in \mathbb{R}^d$, $\boldsymbol{u} \perp \boldsymbol{v}$, the relation holds:

$$g(\boldsymbol{u} + \boldsymbol{v}) = g(\boldsymbol{u} - \boldsymbol{v}). \tag{52}$$

A direct consequence is that for any orthonormal matrix $Q$,

$$
\begin{aligned}
&g(Q(x_1 + \|\boldsymbol{\beta}\|, x_2, \ldots, x_j, \ldots, x_d)) \\
=&g(Q_{-j}(x_1 + \|\boldsymbol{\beta}\|, x_2, \ldots, \overline{x_j}, \ldots, x_d) + Q_j x_j) \\
=&g(Q_{-j}(x_1 + \|\boldsymbol{\beta}\|, x_2, \ldots, \overline{x_j}, \ldots, x_d) - Q_j x_j) \\
=&g(Q(x_1 + \|\boldsymbol{\beta}\|, x_2, \ldots, -x_j, \ldots, x_d)),
\end{aligned}
$$

where $Q_j$ is the submatrix of $Q$ without $j$-th column, and $Q_j$ is the $j$-th column of $Q$. Note that $Q_{-j}(x_1 + \|\boldsymbol{\beta}\|, x_2, \ldots, \overline{x_j}, \ldots, x_d)$ is a vector in the space of column span of $Q_{-j}$ and $Q_j x_j$ is in the linear span of $Q_j$, and they are orthogonal to each other. It is then easy to see that the integrand of the $j$-th coordinate of the LS-EM iterate

$$\mathbb{E}_{\boldsymbol{X} \sim f_{\boldsymbol{\beta}^*} \cdot Q} x_j \tanh(0.5(g(Q(x_1 + \|\boldsymbol{\beta}\|, x_2, \ldots, x_d)) - g(Q(x_1 - \|\boldsymbol{\beta}\|, x_2, \ldots, x_d))))$$

is an odd function in $x_j$, thus vanishes as well. On the other hand, from Lemma H.1, we know that the above generalized condition (52) implies that $g$ function has the same value for $\boldsymbol{x}$ with the same norm, and thus is equivalent to our original assumption.

**Lemma H.1** (Equivalence to Rotation Invariance)**.** *Suppose the condition* (52) *holds. For any* $\boldsymbol{u}_1, \boldsymbol{u}_2$ *satisfying* $\|\boldsymbol{u}_1\| = \|\boldsymbol{u}_2\|$, $g(\boldsymbol{u}_1) = g(\boldsymbol{u}_2)$.

*Proof.* $\|\boldsymbol{u}_1\| = \|\boldsymbol{u}_2\|$ implies that $\frac{\boldsymbol{u}_1 + \boldsymbol{u}_2}{2}$ is orthogonal to $\frac{\boldsymbol{u}_1 - \boldsymbol{u}_2}{2}$. The condition says that

$$g\left(\frac{\boldsymbol{u}_1 + \boldsymbol{u}_2}{2} + \frac{\boldsymbol{u}_1 - \boldsymbol{u}_2}{2}\right) = g\left(\frac{\boldsymbol{u}_1 + \boldsymbol{u}_2}{2} - \frac{\boldsymbol{u}_1 - \boldsymbol{u}_2}{2}\right),$$

and the conclusion follows.    $\square$

## H.2 Assumption of Log-concavity

In 1-D case, we utilize the consistency property to rewrite the difference between the LS-EM update and the ground truth:

$$|M(\beta^*, \beta) - \beta^*| = \left[1 - \frac{\partial M(z, \beta)}{\partial z} \Big|_{z=t\beta^*+(1-t)\beta}\right] |\beta - \beta^*|,$$

for some $t \in (0, 1)$. In particular, it has been shown that

$$\frac{\partial M(z, \beta)}{\partial z} = \underbrace{\mathbb{E}_{X \sim f_z} \tanh\left(0.5 F_\beta(X)\right)}_{T_1}$$

$$+ \underbrace{\mathbb{E}_{X \sim f_z} \left[0.5 X F'_\beta(X) \tanh'\left(0.5 F_\beta(X)\right)\right]}_{T_2},$$

with $T_1, T_2 > 0$. The main property we utilize about the log-concavity is that $F'_\beta(x) \geq 0$ when $g = -\log f$ is convex with non-decreasing derivative. However, when we do not have the convexity, the term $T_2$ can be negative and the factor $\left[1 - \frac{\partial M(z, \beta)}{\partial z} \Big|_{z=t\beta^*+(1-t)\beta}\right]$ can be possibly greater than 1, leading to non-convergence. We demonstrate this behavior in the following simple example: $f \propto \exp(-|x|^{0.25})$. Its non-convergence behavior when the initial iterate $\beta_0$ is in a neighborhood of 0 has been plotted in Figure 1. In Figure 4, we plot the value of $T_1$ and $T_2$ as a function of $z$ to show that the problem occurs due to the negativity of $T_2$.

Figure 4: $f$ is chosen to be proportional to $\exp(-|x|^{0.25})$, $\beta^* = 1$, $\beta = 0.1$, $z \in (\beta, \beta^*)$. We compute $T_1$ and $T_2$ using a finite sample sum with size 1000000. It is seen that $T_2$ is negative and the resulting $T_1 - T_2$ can become ngative.

## H.3 Approximate M-step

We consider the family of polynomial distributions with log density $g \propto |x|^r$ for some $r \geq 1$. In the E-step of the classical EM algorithm, we obtain a lower bound $Q(\cdot \mid \beta)$ for the log-likelihood based on the current estimate $\beta$:

$$Q(b \mid \beta) = \mathbb{E}_{x \sim f_{\beta^*, \sigma}} \left[-p^1_{\beta, \sigma} g\left(\frac{|x - b|}{\sigma}\right) - p^2_{\beta, \sigma} g\left(\frac{|x + b|}{\sigma}\right)\right]. \tag{53}$$

The M-step is to compute $\operatorname{argmax}_b Q(b \mid \beta)$. $Q(\cdot \mid \beta)$ is a concave function, thus the optimization problem has a well-defined solution, however, it does not admit a closed form solution in general. Consider the above example where the ground truth distribution is polynomial, the M-step is equivalent to solving for a polynomial equation with degree $r - 1$.

In the following, we plot the negative $Q$ function (convex) for two polynomial distributions in Figure 5 and Figure 6. Meanwhile we trace two points $(\beta, Q(\beta \mid \beta))$ and $(\beta^+, Q(\beta^+ \mid \beta))$, where $\beta^+$ is the least-squares EM update:

$$\beta^+ = \mathbb{E}_{x \sim f_{\beta^*, \sigma}} \left[x \tanh(0.5 F_{\beta, \sigma}(x))\right].$$

Numerically we find that $b = \beta^+$ strictly increases the value of $Q$ function compared to $b = \beta$ when $\beta$ is not equal to the true parameter.

Figure 5: Plot of $-Q(\cdot \mid \beta)$ for $g \propto |x|$ for $\beta = 0.1, 0.5, 0.8, 1.2, 1.5$. The true parameter $\beta^* = 1$. The blue dots correspond to $b = \beta$ and the red dots correspond to $b = \beta^+$. It is seen that $\beta^+$ is not the exact the M-step, as they do not minimize the $-Q$ function. However, $-Q(\beta^+ \mid \beta) < -Q(\beta \mid \beta)$, suggesting that the least-squares EM update is a type of approximate M-step.

Figure 6: Plot of $-Q(\cdot \mid \beta)$ for $g \propto |x|^{2.5}$ for $\beta = 0.1, 0.5, 0.8, 1.2, 1.5$. The true parameter is $\beta^* = 1$.

## H.4  Non-convergence in $\ell_2$

We provide numerical evidence for Lemma 4.3, which claims the non $\ell_2$ decreasing property for general log-concave distributions. We consider the polynomial family $f \propto \exp(-\|x\|^r)$, for some $r \in [1, \infty)$. We have proved that when $r <= 2$, there is no spurious fixed point in the orthogonal direction. When $r > 2$, there is a spurious orthogonal fixed point in the orthogonal direction.

Figure 7: We plot the LS-EM iterates initialized at $(0, 0.1)$, in the orthogonal direction to the ground truth $\beta^* = (1, 0)$. It is known that the future iterates stay in the orthogonal space. When $r = 1.8$, it is seen that the iterates converge to $(0, 0)$, and when $t = 2.2$, the iterates converge to some non-zero point in the orthogonal direction.

Figure 8: We plot the LS-EM iterates initialized at $(0.0001, 0.1)$, close to the orthogonal direction to the ground truth $\beta^* = (1, 0)$. When $r = 1.8$, the iterates has decreasing $\ell_2$ distance with $\beta^*$. When $r = 2.2$, the $\ell_2$ distance first increases before it decreases.

Figure 9: The ground truth distribution is $g \propto |x|$, with $\beta^* = 1$. The mis-specified distributions are picked with degree $1.5, 2, 3$. We plot $\beta^+$ as a function of $\beta$. The intersection point between $\beta^+$ and $\beta$ is the fixed point for the variant EM update. It is seen that when we fit with a polynomial distribution with higher degree, the fixed points are all greater than 1.

## H.5 Misspecified LS-EM

We consider the family of polynomial distributions with $g \propto |x|^r$ for some $r \geq 1$. For the ground truth distribution, we pick some $r_0$ in the family and fit with another distribution with $r_1$. We observe that when $r_1 > r_0$, the variant EM updates tend to converge to a point greater than $\beta^*$. On the other hand, when $r_1 < r_0$, the variant EM updates tend to converge to a point smaller than $\beta^*$. Numerical evidence can be seen in Figure 9 and Figure 10. In Figure 10, we observe that when fitting a distribution with heavier tail than the ground the truth, the convergence point can be 0, which might lead to a big error in estimation. Therefore, it suggests that one should fit a distribution with a lighter tail in practice. In both sets of experiments, we observe that fitting a 2GMM yields a fixed point close to the ground truth parameter.

Figure 10: The ground truth distribution is $g \propto |x|^2$, with $\beta^* = 1$. The misspecified distributions are picked with degree $1.5, 2, 3$. We plot $\beta^+$ as a function of $\beta$. It is seen that when we fit with a polynomial distribution with higher degree, the fixed points are all greater than $1$. When we fit with a polynomial distribution with lower degree, the fixed points are all smaller than $1$. In particular, when we fit with a Laplace distribution, the only fixed point is $0$.