[Reviews · NeurIPS 2019]

Reviewer 1



The task of demixing a balanced two log-concave densities with the same and known covariance matrix seems restrictive. The authors did not discuss whether the results apply to unbalanced mixtures of two log-concave densities, nor did they discuss the implication of requiring covariances for the two being the same or the knowledge of covariances. The works builds on previous works on global convergence guarantees via EM for a balanced 2 GMM (or mixture of 2 truncated Gaussians) with known covariance, as well as following up works on mixture of 2 linear regressions and mixtures of 2 1-d laplacian distributions. The methodological contribution is to modify the M-step. However, the technical difficulty of proving convergence or providing finite-sample analysis compared with previous works is unclear. The incentive of using EM algorithm is unclear. It is well known that global convergence of EM has been established for mixtures of two densities (not more than two). However, there are other methods such as spectral and tensor methods that are guaranteed to converge to global optima for mixtures of more than two. So the question is why not consider the spectral methods? How does the least squares EM compare with, say tensor methods? The robustness analysis under model mis-specification is under a restrictive setting where the ground truth densities are log-concave and the fitted are Gaussians. I am curious in what happens if the ground truth densities are not log-concave. If the theoretical analysis is difficult to be established, can you show some experimental results? The theoretical analysis seems sound, however it would be nice to show some experimental results. ================ Comments after author response: After carefully reading the author's response, I am more convinced about the technical challenges involved in this paper. I would leave my score unchanged though since I think adding some experiments and a more general model misspecification analysis would make the paper much more impactful.

Reviewer 2



The study of finite mixture of distributions is an important issue in machine learning. Except for the Gaussian Mixture Model (GMM), there is not theoretical guarantee for the parameter estimation problem for parametric mixture models. This paper proposes to provide some guarantee for a wider class of mixture models, namely the mixture of log-concave distributions. The expectation-maximization (EM) algorithm is the standard tool to study mixture models. However, the global convergence of the EM algorithm is not ensured, even in the simple case of GMM. Recently, the global convergence of the EM algorithm has been established for a balanced mixture of two Gaussians, two truncated Gaussians, two linear regressions or two Laplace distributions. The present paper extend these previous works to a mixture of two log-concave distributions. In particular, the explicit formulation of the density is not known in this context. To overcome this issue, the authors propose a new variant of the EM algorithm: the Least Square EM (LS-EM) algorithm. The basic idea is to replace the classical M-step of the EM algorithm by a least square problem. In particular, for GMM, the LS-EM is reduced to the EM algorithm. The authors prove the global convergence of the LS-EM algorithm for a mixture of rotation invariant log-concave distributions, with random initialization. Convergence is established for any dimension in the infinite sampling setting and for dimension one in the finite sampling setting. Comments: This new convergence study is very interesting. However, some assumptions made by the authors reduce the scope of application of this paper. First of all, the covariance of the sampling is supposed known and the estimation concerns only the position parameter. Moreover, the mixture is assumed to be balanced and symmetric. Last, the distributions are assumed to be rotation invariant ; in practice many interesting distributions do not satisfy this condition (as claimed by the authors in their conclusion). Despite these limitations, the authors propose a new way to demonstrate the global convergence of an EM-like algorithm based on a geometric notion of angle-decreasing. They also provide a demonstration of the non-effectiveness of the traditional approach through the l2 distance. The paper is well-written. Unless I am mistaken, the proofs (in the supplementary material) are correct, very clear and easy to follow. There are some typos in the supplementary material but they do not interfere with reading. Numerical experiments are provided in the supplementary material but it is not mentioned in the main text, which is unfortunate. Last, the author refer to a “regular condition” throughout the paper but never explicit it. I guess the authors are referring to the classical assumption for differentiation under the integral sign but I would have liked to see it explained, at least in the additional material.

Reviewer 3



~~~~~~~~~~~ After rebuttal: Thank the authors for clarifying my comments #1 and #3. Since my comments on #2, finite sample analysis for a general dimensional case, was not addressed, I would not increase my rating and would like to keep the current "accept" rating. ~~~~~~~~~~~ This paper studies the theoretical properties in the estimation of location of mixture of two rotation invariant log-concave densities. It shows that a newly proposed LS-EM algorithm converges to the true location parameter with a random initialization. In the finite sample analysis, explicit convergence rates and sample complexity bounds are further developed. Strengths: 1. This paper is very well written and easy to follow. 2. It addresses an important and challenging problems. 3. The theoretical analysis is non-trivial and the technical contribution is high. Weakness: 1. A special case of the proposed model is the Gaussian mixture model. Can the authors discuss the proved convergence rates and sample complexity bounds with that established in GMM (Balakrishnan et al., 2017)? It is interesting to see if there is any accuracy loss by using a different proof technique. Sivaraman Balakrishnan, Martin J. Wainwright, and Bin Yu. Statistical guarantees for the EM algorithm: From population to sample-based analysis. The Annals of Statistics, 45(1):77–120, 2017 2. The finite sample analysis (sample complexity bounds) is only derived for the 1-dimensional case. This largely limits the popularity of the proposed theoretical framework. Can the authors extend the finite sample analysis to a general d-dimensional case, or at least provide some numerical study to show the convergence performance? 3. In Proposition 6.1, the condition \eta \ge C_0 for some constant C_0 seems to be strong. Typically, the signal-to-noise ratio \eta is a small value. It would be great if the authors can further clarify this condition and compare it with that in Section 4 (correct model case).

[Author Response · NeurIPS 2019]

We thank the reviewers for their insightful comments. Below we prioritize what we view as the most important points.

**Motivation and Contributions**

EM is the quintessential approach for mixture problems. Despite its long history and popularity, theoretical under-
standing of EM is disappointingly limited and largely lags its empirical application. Even in the simplest setting of 2
Gaussians (2GMM), global convergence was only established recently. We establish such a guarantee for a much more
general class of distributions. In particular, *we identify the crucial role of log-concavity—rather than Gaussianity—in*
*ensuring global convergence*, while recent related work considers specific distributions (e.g. Laplace or regression with
Gaussian noise) on a case-by-case basis.

Compared to prior work, we overcame two main technical challenges: 1) We need to establish the angle shrinkage
property of the LS-EM algorithm. This is contrastingly different from the working mechanism (shrinkage in distance)
of classical EM for 2GMM; existing work (e.g. Balakrishnan, and Daskalakis) heavily relies on this mechanism, which
does *not* work for general log-concave mixtures. 2) Unlike Gaussian, for general log-concave distributions in the high
dimension, the coordinates are dependent of each other even when the covariance matrix is identity; therefore, a more
sophisticated sensitivity analysis (Lemma D.1) is needed to establish angle shrinkage.

Compared to tensor methods, EM is much simpler, and manifests *linear* dependency on the dimension $d$ in terms of
time and sample complexities (as opposed to polynomial for tensor methods).

**Assumptions**

As Reviewer 1 pointed out, our analysis is built upon several assumptions (a balanced mixture of 2 distributions with
same covariance). We note that these assumptions are common among recent literature on global convergence of EM.
Moreover, there exist examples of more general mixtures where global convergence is impossible, including mixture of
3 Gaussians (see arXiv 1609.00978) or 2 unbalanced Gaussians (see arXiv 1810.11344). Additional comments below:

(i) Log-concavity is crucial. In our analysis, it guarantees that certain
derivatives of the LS-EM operator are non-decreasing.

In our experiment, if the ground-truth distribution is not log-
concave, with symmetric density function $\frac{1}{2}1_{0<x\le0.5}+\frac{1}{8}1_{0.5<x<1.5}+$
$\frac{1}{32}1_{1.5\le x<3.5}+\frac{1}{128}1_{3.5\le x<7.5}+\frac{1}{512}1_{7.5\le x<15.5}$ (defined symmetri-
cally on the negative side), then LS-EM *incorrectly* converges to 0
when the initial solution $\beta_0$ is close to 0 (see figure on the right).

(ii) For model misspecification, we focus on fitting with Gaussian, as
it is a typical and reasonable choice in practice. Experiments show
that if the fitted distribution is not Gaussian, LS-EM may fail completely (Fig. 6 in Appendix of original paper).

(iii) The assumption of symmetric parameters $\pm\beta^*$ is just a form of centering and hence *not* essential to our results.

It is definitely an intriguing problem to figure out the exact setting in which global convergence can be achieved, or how
one can modify the standard EM algorithm to avoid spurious fixed points. Recent work in arXiv 1810.11344 demon-
strated that over-paramerized EM[1] converges globally for unbalanced (but still symmetric) 2GMM. By considering
general log concave distributions, we view our work as another step towards a more complete theory.

**Experiments**

We totally agree on the importance of experiments. We did present numerical experiments for i) robustness under model
misspecification, and ii) connection between LS-EM and classical EM, in the appendix due to page limit. Additional
experiments (including the one above), which will be included in the final version, corroborate our theoretical findings
and further explore the issues of model misspecification and sensitivity to the assumptions, as reviewer 1 suggested.

**Other Comments**

**Comparison with Balakrishnan's paper [BWY17]:** When specialized to 2GMM in 1-D, our results match those in
BWY17. For higher dimensions, we establish global convergence (the analysis in BWY17 is local); while we currently
do not have explicit bounds on the convergence rate and sample complexity, we expect they would again match BWY17.

**Regularity Conditions:** These conditions are explicitly explained in Section E of the appendix. They are indeed about
differentiation under integral sign (as pointed out by Reviewer 2), and are satisfied by common log-concave mixtures.

**SNR Condition in Proposition 6.1:** Reviewer 3 mentioned that this condition is strong. Note that Proposition 6.1 is a
simplified version of the more general result established in the proof of the proposition (pp. 32). There we derived the
following bound for the distance between the misspecified solution and the true solution: $|\overline{\beta}-\beta^*|\le\frac{9\sigma}{1-\exp(-0.125\eta^2)}$.
This bound holds for any SNR $\eta>0$; when the SNR is smaller, the bound becomes worse, as natural.

## Footnotes

[1]It considers the weight parameter as a variable in the EM algorithm, even though the weight is known apriori


[Meta-Review · NeurIPS 2019]

This paper analyzes least square EM for mixture of two log concave distributions (that are rotationally invariant) and proves global convergence. This generalizes previous results on mixture of two Gaussians. Reviewers found the paper to be reasonable, although there is also doubt on the applicability given that the result cannot be generalized to 3 or more components.